

# Potential for Equation Discovery with AI in the Climate Sciences

Chris Huntingford[1], Andrew J. Nicoll[2], Cornelia Klein[1], and Jawairia A. Ahmad[1]

[1]U.K. Centre for Ecology and Hydrology, Benson Lane, Wallingford, Oxfordshire, OX10 8BB, U.K.
[2]Atmospheric, Oceanic and Planetary Physics, Clarendon Laboratory, Oxford, OX1 3PU, U.K.

*Correspondence to:* Chris Huntingford (chg@ceh.ac.uk)

**Abstract.**

Climate change and Artificial Intelligence (AI) are both attracting great interest across society. There is also substantial interest in merging the two sciences, with evidence already that AI can identify earlier precursors to extreme weather events. There are a range of AI algorithms, and selection of the most appropriate one maximizes the amount of additional understanding extractable for any dataset. However, most AI algorithms are statistically based and even with careful splitting between data for training and testing, they arguably remain as emulators. Emulators may make unreliable predictions when driven by out-of-sample forcing and climate change is an example of this, requiring understanding responses to atmospheric Greenhouse Gas (GHG) concentrations that may be substantially higher than present or the recent past. Notable, though, is the emerging AI technique of "equation discovery". AI-derived equations from data also does not automatically guarantee good performance for new forcing regimes. However, access to equations rather than a statistical emulator guides system understanding, as their variables and parameters often have a better interpretation. Better process knowledge enables judgements as to whether equations are trusted under extrapolation. For many climate system attributes, descriptive equations are not yet fully available or may be unreliable. This uncertainty is hindering the development of Earth System Models (ESMs) which remain the main tool for projections of large-scale environmental change as GHGs rise. Here, we make the case for using AI-driven equation discovery in climate research, given that its outputs are more interpretable in terms of processes. As ESMs are based around the numerical discretisation of equations that describe climate components, equation discovery from new datasets provides a format amenable to direct inclusion into such models where representation of environmental systems is missing. We present three illustrative examples of how AI-led equation discovery may advance future climate science research. These are generating new equations related to atmospheric convection, parameter derivation for existing equations of the terrestrial carbon cycle, and (additional to ESM improvement) the creation of simplified models of large-scale oceanic features to assess Tipping Point (TP) risks.

## 1 Introduction

Addressing climate change due to human burning of fossil fuels remains a three-fold challenge for society and science. The first challenge is to determine what constitutes a broad "safe" maximum level of global warming, for which there are already proposals of 1.5°C or 2.0°C (UNFCCC, 2015) above preindustrial times. One guide is to constrain global warming to levels that avoid trigger large-scale TPs (e.g. Abrams et al., 2023) (or even a self-perpetuating cascade of TPs; Wunderling et al.



(2021)), where major changes occur to Earth system components for small extra temperature rises. Once a warming threshold is adopted, the second requirement is to support adaptation planning by determining detailed local changes in near-surface meteorology corresponding to that global temperature rise. The third task is to derive GHG emissions profiles compatible with eventual stabilisation of global warming at prescribed target levels. Knowledge of such profiles encourages mitigation

plans to develop technologies for a transition from using fossil fuels sufficiently fast to prevent key global warming threshold exceedence. All three activities depend on accurate projections of changes to the climate system and the global carbon cycle in response to rising GHGs. Current understanding of such expected changes is presented in reports by the Intergovernmental Panel on Climate Change (IPCC), of which the latest is the sixth assessment (IPCC, 2021). However, this latest report highlights substantial remaining uncertainties in key climate components. Such uncertainties aggregate, affecting the constraining of

summary global parameters such as Equilibrium Climate Sensitivity (ECS), which is global warming in a stabilised climate for doubling of atmospheric $CO_2$ concentrations. The range of ECS values estimated by ESMs remains substantial (Forster et al., 2021). Regionally for many locations, there remains major uncertainty in how hourly to annual rainfall levels will change as GHGs rise (e.g. Tebaldi et al., 2021), including for extremes (Lenderink et al., 2017; Lenderink and Fowler, 2017). Uncertainty in ECS value makes mitigation planning difficult, leading to poor knowledge of the reductions of $CO_2$ emissions needed to

keep global warming below a target such as two degrees. Uncertainty in future changes to rainfall statistics prevents adaptation planning for adjustments to future flood or drought frequencies.

Earth System Models (ESMs) are large computer codes designed to estimate climate change for different prescribed trajectories of future atmospheric GHG concentrations or emissions. The basis of ESMs is the numerical discretisation (at scales of typically 100km) of equations that describe all Earth system features, including the oceans, land surface, atmosphere and

cryosphere, and their feedbacks. Analysis of ESM diagnostics has enabled breakthroughs in climate system understanding, and a particular community achievement is that approximately twenty research centres contribute model output to a common database, available for analysis by researchers. The latest ensemble of models is the Coupled Model Intercomparison Project version 6 (Eyring et al., 2016). However, the large uncertainties noted above are derived from differences between ESMs. Hence, a key requirement for climate researchers is to understand and remove such differences, to create refined projections

with smaller uncertainty bounds. An interim approach to uncertainty reduction is the method of emergent constraints (e.g. Hall et al., 2019; Williamson et al., 2021; Huntingford et al., 2023), which searches for inter-ESM regressions between quantities that are also measured and changes of importance in the future. Where robust regressions are found, measurements use this to constrain the future quantity. However, while emergent constraints provide a powerful methodology to lower inter-ESM spread, ultimately needed are ESMs that, for some remaining processes, have improved equation representation with accurate

parameterisation.

As climate science has progressed through ESMs development in recent decades, so have Artificial Intelligence (AI) algorithms. The potential applications of AI in society are vast and include opportunities to advance scientific discovery (e.g. Wang et al., 2023). As expected, there are calls for AI to be applied to climate science (Jones, 2017) and in detail (e.g. Schneider et al., 2017; Huntingford et al., 2019; Reichstein et al., 2019; Eyring et al., 2024). Already AI has been found to have a

strong ability to alert to emerging extreme climate events (e.g. Bi et al., 2023; Lam et al., 2023), and the timing or onset of





oscillatory features of the climate system, such as the Madden-Julian Oscillation (MJO) (Delaunay and Christensen, 2022). Yet most AI algorithms are statistically based and so there is growing interest in applying newer physics-informed methods (Karniadakis et al., 2021) to support understanding climate system components Kashinath et al. (2021). Physics-informed approaches ensure that AI-derived findings retain at least some consistency with known underlying process differential equations.

Examples of applications include the reconstruction of atmospheric properties of tropical cyclone events (Eusebi et al., 2024) and characteristics of extreme precipitation (Kodra et al., 2020).

Even more recently, a branch of AI has emerged termed "AI-led equation discovery", which derives candidates for the governing equations that describe any dataset under investigation. Unlike physics-led approaches, the technique instead uses AI to discover hereto unknown equations. The authors who initially suggested this possibility include (Raissi et al., 2019, their

Section 4) as well as Champion et al. (2019); Brunton et al. (2016); Rudy et al. (2017). As suggested above, the advancement of ESMs implies the development of the equations encoded in them. Hence, we now consider how this form of AI may support ESMs by discovering any required missing equations and parameters.

## 2   AI Methods and Including for Equation Discovery

### 2.1   Background climate analysis methods and existing AI methods

The range of AI methods is vast and the correct one to apply depends on the issue being investigated in the data such as frequency, spatial size, level of system nonlinearity, and if there are "labels" describing the effects being searched for. The latter point determines whether to use supervised or unsupervised algorithms. The development of ESMs has traditionally been driven by advances in contemporary knowledge of geophysical processes and related mathematical models. However, climate research has also been influenced by statistically based research, some of which may be regarded as precursors of

more modern AI techniques. In this Section we give: (1) a brief overview of some traditional statistical approaches to climate analysis, (2) describe the application of generic and currently available AI algorithms to climate science, then (3) review currently available AI algorithms in a more general non-climate context and (4) consider the newer techniques including physics-informed calculations, again for the broader application background. To achieve this summary and for each of these four points, we point to and make brief summaries of four influential textbooks (Figure 1). For an initial statistical analysis

of climate attributes, we select "keywords" from some section headings of Storch and Zwiers (1999). Early applications of machine learning applied to environmental issues, including forecasting and components of the climate system, are presented in Hsieh (2009). For a general but extensive overview of available machine learning algorithms, we use Murphy (2013). Moving to the main theme of this perspective, Brunton and Kutz (2022) summarises very current methods of data-driven machine learning, including physics-led techniques.

In more detail, Storch and Zwiers (1999) describe the initial application of statistical methods to climate-related research, including probability theory, timeseries analysis, Eigen techniques and Empirical Orthogonal Functions (EOFs). The EOF method is popular for spatiotemporal analyses of physical climate variables (Smith et al., 1996; Mu et al., 2004; Hannachi et al., 2007). EOFs reduce the degrees of freedom of key variables (such as Sea Surface Temperatures; Smith et al. (1996)), and





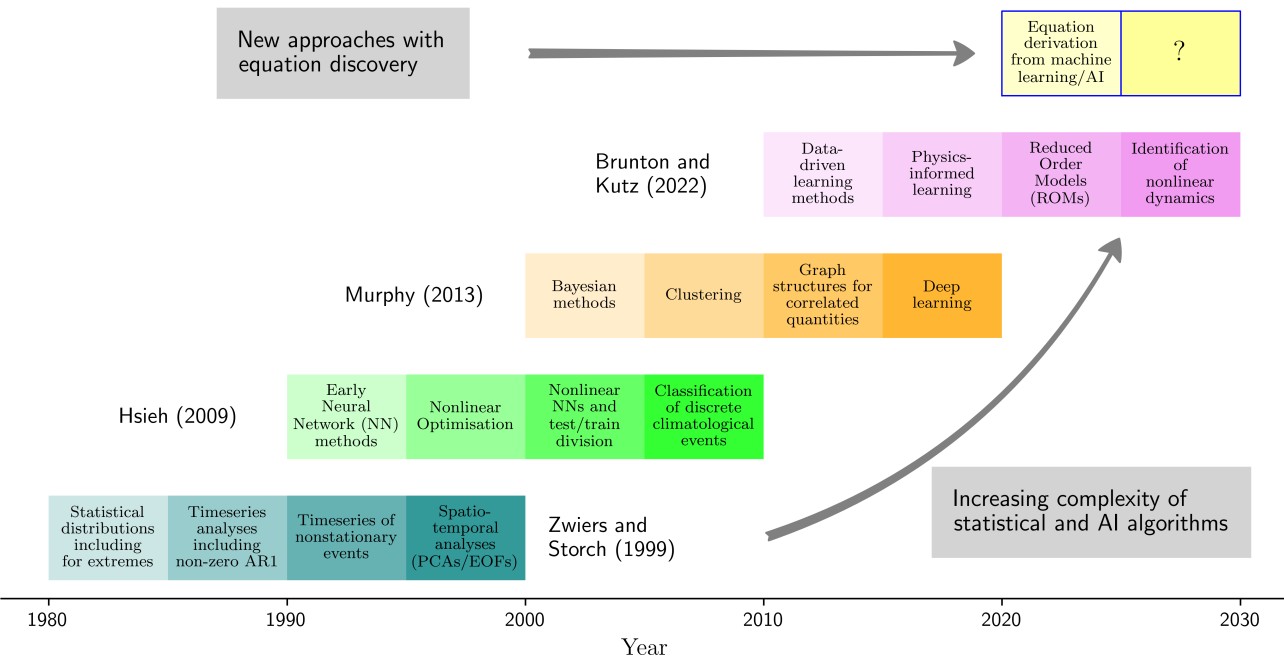

**Figure 1.** Schematic illustrating the evolution of the application of statistical methods to climate research, as well as more recent general developments in AI methods. The techniques illustrated and applied to environmental research are based on sections of the books by Storch and Zwiers (1999) and Hsieh (2009). More generic AI developments, not necessarily used in climate research, are linked to parts of the books by Murphy (2013) and Brunton and Kutz (2022), where the latter considers physics-led algorithms. The top bar suggests using recent advances in AI that are capable of deriving underlying process equations to better determine features of the climate system where uncertainty remains. The call for applying AI-led equation discovery to climate research is the main subject of this commentary. We retain the idea that AI may also support climate research in ways not yet considered, as shown by the question mark

are often presented as spatial patterns capturing geographical modes of variability multiplied by timeseries of their magnitudes. EOFs enable a simpler way to characterise climate models and therefore allow easier comparison against gridded datasets (e.g. Mu et al., 2004). Early neural networks, inspired by how the human brain is believed to operate, evolved from the perceptron model to hidden-layer models (Hsieh, 2009). Standard multivariate regression and EOF methods contain strong

5   implicit assumptions of linearity, while neural networks in all forms contain nonlinear elements. Also of importance is the widespread application of Bayesian statistics in climate science. Bayesian statistics provide information on state variables that also include *a-priori* knowledge about such variables. Bayesian methods, therefore, hint at the newer physics-led approaches. The application of deriving Bayesian probability distributions for climate quantities matured during the first decade of the 21st century. For example, Berliner et al. (2000) employ such methods to the problem of detection and attribution of human

10  forcing of the climate system, as represented by near-surface temperature fields. Boulanger et al. (2006) determine the dominant





features of the temperature variability in South American data, which is then used to compare to the performance of the ESMs and thus weight such modes and hence their future projections. Similar Bayesian-based analyses for near-surface temperature but for multiple regions across the globe are performed by Smith et al. (2009).

Most early studies with AI elements focus on supervised learning, where the training is performed using labelled target
data. However, in recent decades, clustering algorithms have substantially increased the popularity of unsupervised learning. Unsupervised learning, by definition, raises the exciting possibility of algorithms that interpret climate data or ESM-based outputs in new ways. The broad analysis of Steinbach et al. (2003) uses clustering to identify climate indices that characterise many behaviours of the oceans and the atmosphere. Similarly, Lund and Li (2009) use clustering of autocorrelated climate timeseries to identify the areal bounds of distinct climate zones. Graphical models, also an unsupervised method, provide a
novel technique to represent linked relationships between variables of interest. Ebert-Uphoff and Deng (2012) use graphical models to explore causal relationships between atmospheric circulations and provide a framework with potential to discover further causal links between climate variables.

Many contemporary scientific problems, including climate research, involve interpretating exceptionally large datasets. The advent of "Big Data" methods has fuelled a branch of analysis of climate models and data that generates better forecasting
methods, supporting enhanced or earlier warnings of extreme events. For example, Liu et al. (2008) compared different data-driven learning methods to downscale weather forecasts, to provide statistics of near-surface meteorological conditions at single points or very small spatial scales, and which may include local predictions of extremes. Additionally, the climate system contains strong and complex nonlinear interactions that operate over multiple timescales including that of forecasting. Yet, despite this complexity, there may be robust underlying reduced-complexity nonlinear dynamic systems awaiting discovery. Tradi-
tional scale analysis of underlying equations may reveal such dynamical systems, including the famous paper of Lorenz (1963) which is a three-variable system of coupled ordinary differential equations that simulate aspects of atmospheric convection. The suggestion is that newer algorithms may routinely identify the dominant processes in a complex system, and much hope is placed in the Long-Short Term Memory (LSTM) technique (which is a recurrent neural network algorithm; Vlachas et al. (2018)), designed to forecast high-dimensional chaotic systems. Recent studies also highlight the performance benefits of deep
learning methods that analyse key datasets to improve medium-range weather forecasting (Bi et al., 2023; Lam et al., 2023). Although these two examples are for much shorter timescales than explaining emerging climatic signals on decade-to-century timescales, they illustrate the usefulness of AI in extracting additional information from the complex atmospheric system.

Reduced Order Models (ROMs) project partial differential equations (PDEs) such that the dominant dynamic processes are encoded in low-rank spaces. The method allows for improvements in computational speed and optimisation of any full
parameterised PDE system. ROMs can effectively project PDE dynamics to low dimensional spaces so simulations of the full PDE model can be better evaluated and understood. Proper Orthogonal Decomposition (POD) is a key method for creating a ROM, is used to study complex spatiotemporally dynamic systems in fluid dynamics (e.g. for oceanic circulations; San and Iliescu, 2015) and provides a viable way to interpret ESM diagnostics. EOFs are an earlier form and a subset of ROMs, but we suggest that the full utility of ROMs and their many potential configurations and applications is still largely unexplored





for climate science. Brunton and Kutz (2022) make the case that, ideally, a process interpretation is derived for the main components of any decomposition.

Yet despite the remarkable progress with creating AI methods, most are highly statistical in construction. There is a growing view (again reflecting the Bayesian viewpoint) that additionally, AI needs to recognise that there are underlying processes

and for which there often exists substantial information and knowledge. Hence, physics-informed learning is gaining traction, providing methods of constraining machine learning-based predictions using laws of physical rationality. Karniadakis et al. (2021) reviewed recent trends in this approach by embedding physics in machine learning and concluded that combined data and physics-based model integration can be achieved even in uncertain and high-dimensional contexts. That research discussed several applications of physics-informed learning for inverse and ill-posed problems in fluid dynamics, thermodynamics, and

seismology, illustrating the possibility of increased process consistency but expressed via neural network architectures.

We now turn to what we suggest as the next frontier in AI developments and of substantial potential use to climate science. This next step is for where there is uncertainty in the underlying physical processes, AI derives the underlying descriptive equations. Such a discovery can constitute either the full equation set or a smaller reduced-complexity set as a dynamical system that captures the dominant system responses. The upper row of Figure 1 captures this as an emerging direction for

machine learning (ML) or AI (we use the terminology of ML and AI interchangeably; see Kuehl et al. (2022) for precise definitions and how the two differ).

Equation discovery using machine learning is well positioned to advance our understanding of Earth's climate that contains nonlinear features, given the basis of much AI is to find underlying nonlinearities. A specific approach is symbolic regression and is the most common AI-based approach to discover equations implicit in data. This form of regression procedure searches

a space of mathematical expressions to find the optimal combination (i.e. a symbolic model) that best fits the data. Sparse regression is a type of symbolic regression method which has the advantage of diminishing the search space of possible terms in the equation discovery process, substantially reducing the likelihood of over-fitting to the observed data. Brunton and Kutz (2022) place a special emphasis on a sparse regression method, which they refer to as the Sparse Identification of Nonlinear Dynamics (SINDy), as a data-driven approach to uncover reduced order models (ROMs) of systems with unknown spatial-

temporal dynamics.

## 2.2  Symbolic Regression Methods for Equation Discovery to Uncover Unknown Dynamics

Equation discovery techniques can be categorised as data-driven or knowledge-driven discovery (Tanevski et al., 2020). These approaches involve inferring the best possible derived model structure and parameter values by ensuring minimal error between observations and model predictions. The former approach, considered to be general AI-led equation discovery, is applicable for

systems where there is very little or no understanding of the underlying dynamics and, therefore, no obvious model structure preexists. The latter approach relies on existing expert knowledge of the system, in which those developing the discovery process ensure features of the existing models remain in the new derived equations.

Where the behaviours of a dynamical system are largely or completely unknown, an emerging method to determine the underlying equations is that of symbolic regression. The data-driven symbolic regression algorithm does not depend on user-





specified prior knowledge of a system. Hence, unlike a usual regression task that involves a predefined model structure, symbolic regression finds the optimal model and its parameters that best fit the data.

The usual form of symbolic regression, which can effectively minimise both model complexity and prediction error, is sparse regression, which is the main focus of this section. However, we first note other methods, such as a deep learning-based symbolic regression model proposed by Petersen (2019) that uses a recurrent neural network with a "risk-seeking policy gradient" to generate better fitting expressions. This approach has been shown to be robust against noisy data. Another type of symbolic regression method is that of Genetic Algorithms (GA) (Keren et al., 2023). GAs can include prior physical knowledge of the system in the optimisation procedure, and works particularly well for systems with strong linearity. This technique involves building "trees" of random symbolic expressions and using stochastic optimisation to perform the replacement and recombination of tree subsamples. Ultimately, this finds the combination of terms that best fit the data.

Common to these three symbolic regression methods (sparse regression, deep learning and GAs) is an optimisation procedure which finds a linear combination of (potentially nonlinear) functions from a large functional space which best fits the underlying system behaviour. The quickest and most general approach is to use sparse regression, which substantially reduces the search space of possible functions. Such speed is needed, compared to a computationally inefficient "brute-force" method of looping over all combinations of possible contributing functions. Sparse regression also reduces the likelihood of over fitting, generating equations with limited terms, although sufficient to explain the features of the underlying datasets. A popular sparse regression algorithm developed by Brunton et al. (2016), known as Sparse Identification of Nonlinear Dynamics (SINDy), identifies the simplest (parsimonious) model that describes the dynamics of nonlinear systems implicit in data. SINDy investigates time series data to extract interpretable and generalisable models in the form of ordinary differential equations evolving in time. In the event of multiple timeseries spanning a spatial region, then SINDy can determine partial differential equations. A general dynamical system model takes the form of $\dot{\boldsymbol{x}}(t) = \boldsymbol{f}(\boldsymbol{x}(t))$ where the vector $\boldsymbol{x} = [x_1(t) \quad x_2(t) \quad \cdots \quad x_d(t)]^{\mathrm{T}} \in \mathbb{R}^d$ represents the state of the system at a single time instance, $t$, consisting of $d$ system variables. The SINDy algorithm finds a function $\boldsymbol{f} : \mathbb{R}^d \to \mathbb{R}^d$ defining the dynamics and time evolution of the system. Collecting a time-history of the state $\dot{\boldsymbol{x}}(t)$ across the $m$ set of times $t_1, t_2, ..., t_m$ produces the complete $m \times d$ data matrix,

$$
\mathbf{X} = \begin{bmatrix} \boldsymbol{x}^{\mathrm{T}}(t_1) \\ \boldsymbol{x}^{\mathrm{T}}(t_2) \\ \vdots \\ \boldsymbol{x}^{\mathrm{T}}(t_m) \end{bmatrix} = \begin{bmatrix} x_1(t_1) & x_2(t_1) & \cdots & x_d(t_1) \\ x_1(t_2) & x_2(t_2) & \cdots & x_d(t_2) \\ \vdots & \vdots & \ddots & \vdots \\ x_1(t_m) & x_2(t_m) & \cdots & x_d(t_m) \end{bmatrix} \tag{1}
$$

The symbolic regression task is to find the form of $\boldsymbol{f}$ from a time series of the state $\mathbf{X}(t)$ that maps to the derivative $\dot{\mathbf{X}}(t)$, and that is valid across the $m$ set of times $t_1, t_2, ..., t_m$ at which data is available. In order to find a sparse representation of $\boldsymbol{f}$, an augmented library, we first start with $\boldsymbol{\Theta}(\mathbf{X})$, consisting of $n$ candidate functions. The individual functions contributing to $\boldsymbol{f}$ may include polynomial and trigonometric terms. This construction gives a library of dimensions $n \times m \times d$. We show this construction below, where the horizontal direction (size $n$) are the candidate functions, the vertical direction (size $m$) are





the timesteps, and "out of the page" is size $d$ which are the different state variables. In the matrix $\boldsymbol{\Theta}(\mathbf{X})$ below, functions can include "cross terms", so for instance a quadratic term $\mathbf{X}^2$ and for $d = 2$ would have $x_1^2$, $x_2^2$ and additionally $x_1 x_2$ terms (see Eqn. 2 of Brunton et al. (2016)).

$$\boldsymbol{\Theta}(\mathbf{X}) = \begin{bmatrix} | & | & | & | & & | & | & | & | & \\ 1 & \mathbf{X} & \mathbf{X}^2 & \mathbf{X}^3 & \cdots & \sin \mathbf{X} & \cos \mathbf{X} & \sin 2\mathbf{X} & \cos 2\mathbf{X} & \cdots \\ | & | & | & | & & | & | & | & | & \end{bmatrix}. \tag{2}$$

The sparse regression problem is then set up as $\dot{\mathbf{X}}(t) = \boldsymbol{\Theta}(\mathbf{X})\boldsymbol{\Xi}$ where we want to solve for the matrix $\boldsymbol{\Xi} \in \mathbb{R}^{nd}$ which contains vectors of $n$ coefficients corresponding to the linear expansion for each of the $d$ state variables, $\boldsymbol{\Xi} = \begin{bmatrix} \boldsymbol{\xi}_1 & \boldsymbol{\xi}_2 & \cdots & \boldsymbol{\xi}_d \end{bmatrix}$. For each state variable, the fitting procedure attempts to minimise the difference between $\mathbf{y}$ and $\boldsymbol{\Theta}\boldsymbol{\xi}$ where $\mathbf{y}$ is a vector of $m$ data measurements (i.e. a column of $\mathbf{X}$). However, this minimisation is a single sweep across all state variables, and so is not the best fit for each individual variables.

Various sparse regression optimizers can solve for $\boldsymbol{\xi}$. A common algorithm known as LASSO introduces sparsity to the regression procedure via an $L^1$ regularisation term: $\boldsymbol{\xi} = \operatorname*{argmin}_{\boldsymbol{\xi}'} \left\| \boldsymbol{\Theta}\boldsymbol{\xi}' - \mathbf{y} \right\|_2 + \lambda \left\| \boldsymbol{\xi}' \right\|_1$. The key result of solving for $\boldsymbol{\Xi}$ using sparse regression is the coefficient vectors that it obtains are sparse (where most entries are set to zero) due to the optimisation procedure. This means that only a few nonlinear terms in the candidate library are active and therefore included in the right-hand side of one of the row equations $\dot{\boldsymbol{x}}_k = \boldsymbol{f}_k(\boldsymbol{x})$. This leads to a sparse representation of $\boldsymbol{f}$ and therefore parsimonious dynamical

models.

A particularly comprehensive verification of the capability of sparse identification to derive equations is presented in Chen et al. (2021). In that analysis, and pretending to have no knowledge beforehand of the underlying equations, five fundamental governing equations are reproduced and purely from data. These equations are those of Burgers, Kuramoto-Sivashinsky, non-linear Schrödinger, Navier-Stokes and a reaction-diffusion equation. Although we place an emphasis in this section on sparse

regression for equation discovery where we might have little or no knowledge of underlying model structures beforehand, this method may also be used where there exists some process understanding. Such an application is closer to a physics-informed approach. In these circumstances, the library $\boldsymbol{\Theta}$ is restricted to take only a limited set of functional forms based on such *a priori* knowledge, possibly allowing faster convergence of the optimisation procedure as some components of the dynamical system are known.

The SINDy algorithm can additionally include forcing variables in the sparse representation of the dynamics, known as "SINDy with control" (Brunton et al., 2016). This configuration gives the ability to simultaneously disambiguate the internal dynamics of a system and the effect of forcing variables. For climate modeling, an external forcing variable could be a time-series of GHG emissions, their atmospheric concentration levels, or radiative forcing that integrates the effect of all changes in different GHG concentrations. One of the important properties of dynamical systems is stability, which is not guaranteed with

the standard SINDy regression algorithm. For physical systems involving fluid flows where the underlying equations are known to be energy-preserving, although also nonlinear (e.g. having quadratic terms), the "Trapping SINDy" algorithm is available, based on the Schlegel–Noack trapping theorem (Kaptanoglu et al., 2021). This algorithm offers necessary conditions for the





discovered models to be globally stable and energy conserving. We note that the confirmation of basic conservation proper-
ties is a cornerstone of ESM development and testing. The SINDy algorithm was originally used to only discover systems
of ordinary differential equations (ODEs) but was quickly extended to search for partial differential equations (PDEs), using
an algorithm known as "PDEFIND", which fully captures the spatial-temporal behaviour of dynamical systems (Rudy et al.,
5   2017).

There are computer packages that implement the SINDy algorithm and its configurations (for example, trapping capability),
for ODE and PDE systems, such as the Python-based PySINDy package (Brunton et al., 2016).

## 3   Potential Applications of AI-led Equation Discovery

We discuss the potential application of AI-led equation discovery to three Earth system components. In each example, there is
presently a deficiency in understanding, causing uncertainty in the representation of processes by equations. Each application
falls into one of three categories.

In the first example, we address the requirement to better parameterise small-scale convective events at the larger scale to
enable planet-wide representation in coarser-scale Earth System Models. In this instance, arguably, we do not understand the
form of the governing equations.

In the second example, we consider closing the global carbon cycle, where the largest uncertainty is often the magnitude of
atmosphere-land $CO_2$ exchanges. We suggest seeking parameters valid at the large ESM gridbox scale, although for placement
in existing land equations. Due to parameter uncertainties, global land-atmosphere $CO_2$ exchange is often derived as a residual,
after contemporary $CO_2$ emissions and changes in atmospheric and oceanic carbon content are accounted for (e.g. Canadell
et al., 2007), circumnavigating using a land surface model. However, while this provides valuable contemporary information,
it prevents predictions of future land changes. Land surface models are improving (Blyth et al., 2021) with new key processes
already represented by equations, but their parameterisation may apply only at the field scale or smaller, depending on data
used for calibration. Yet the land surface is heterogeneous, providing an opportunity for algorithms to determine equation
parameters that instead aggregate fine-scale processes to ESM gridbox scale. In some instances, terrestrial processes do remain
poorly understood, and so equation discovery may also identify additional equation terms that capture such effects. Hence we
focus on whether AI may advance existing equations by deriving parameters valid at large spatial scales, but note discovery
methods might also characterise missing processes in equation form.

Our third example concerns ocean circulations where the governing equations are fully understood at the local scale, but of
interest is how their internal interactions aggregate to create regional and global responses. Spatially upscaled computationally
fast equations generate key knowledge of oceanic response for a broad range of potential future GHG trajectories, for which
ESMs have not simulated. Many reduced complexity large scale ocean models exist but the equations are presently estimates.
We conjecture that AI-generated spatial aggregation may refine such equations. An additional benefit is that comparing these
simpler models with large-scale oceanic datasets may provide information on the performance of ESMs from which AI has
derived the large scale equations.



### 3.1 Large-scale Parameterisation of Fine Resolution Convective Events

The representation of convection remains a major shortcoming in traditional ESMs, where grid scales of 50-200 km cannot explicitly resolve convection, necessitating parameterisation. These empirical parameterisations simulate the effect of sub-grid vertical displacement of mass, energy, and water on the ESM gridbox scale, producing modelled rainfall as a result. However,

common convective parameterisations often fail to capture typical diurnal cycles of cloud cover and rainfall (Fosser et al., 2015; Prein et al., 2013), with too frequent and light rainfall estimates. Such parameterisations also struggle to represent long-lived convection that propagates across multiple gridboxes, organizing the atmosphere on the mesoscale (Stephens et al., 2010). Meanwhile, rainfall intensities are rising with global temperatures, scaling with the water-holding capacity of a warmer atmosphere at 7% $K^{-1}$ on average, following the Clausius-Clapeyron relationship (e.g. Westra et al., 2014). However, this

statistic does not account for complex meso-scale dynamics unresolved by ESMs. Thus, shortcomings in sub-grid convection representation in ESMs have significant implications for climate change preparedness, limiting the reliability of future rainfall intensification estimates.

    Convection is complex, with governing equations not amenable to direct analytical analysis. Therefore, the current approach involves discretising these equations and conducting Convection-Permitting (CP) simulations on high-resolution ( <10 km)

model grids, which better represent convective storms (Kendon et al., 2017) (c.f. Fig 2). Unfortunately, due to the high computational requirements of fine-resolution calculations, global climate CP simulations have yet to emerge, remaining flagship proposals by only a few major computing centres. Consequently, CP climate simulations are currently limited to specific spatial domains and time periods (Kendon et al., 2021; Stevens et al., 2019).

    Nevertheless, these individual simulations enable us to assess the added value of high-resolution global climate projections,

particularly by comparing CP simulations with diagnostics from lower-resolution climate models (e.g. Fosser et al., 2024). However, due to the small number of simulations and their limited temporal extent, CP models currently provide little information on projection uncertainty or transient climate behaviour. An alternative is to perform multiple CP calculations in parallel for specific target regions, using ESM boundary conditions. Such limited-area downscaling provides valuable regional information but prevents the modelling of large-scale feedbacks ('upscale effects'), which are expected to change and modulate

how climate evolves as GHGs rise.

    A key challenge for climate science is to derive mean large-scale governing equations that accurately present the local statistical properties of convective storms. These equations need to simulate how storm properties will respond to higher levels of GHGs and, crucially, how any changes feed back to the large-scale climate system. Thus, such mean calculations must be designed for integration into coarse-scale ESMs. A promising strategy is to use AI to analyse available CP simulations,

treating these as 'true data' despite being computer-generated (Rasp et al., 2018). Processes that could be extracted from CP models to enhance ESM convective parameterisations include the interaction of storm-scale circulations with large-scale wind, temperature, and humidity fields O'Gorman and Dwyer (2018), the effects of convective upscale growth (Bao et al., 2024), entrainment variability due to wind shear (Mulholland et al., 2021), and the relative importance of thermodynamic versus dynamical drivers of precipitation changes under global warming (Klein et al., 2021). Key target variables would include



gridbox-mean temperature, humidity, and momentum for direct ESM use, as well as cloud cover properties (Grundner et al., 2024) and distributions of convective precipitation and extremes.

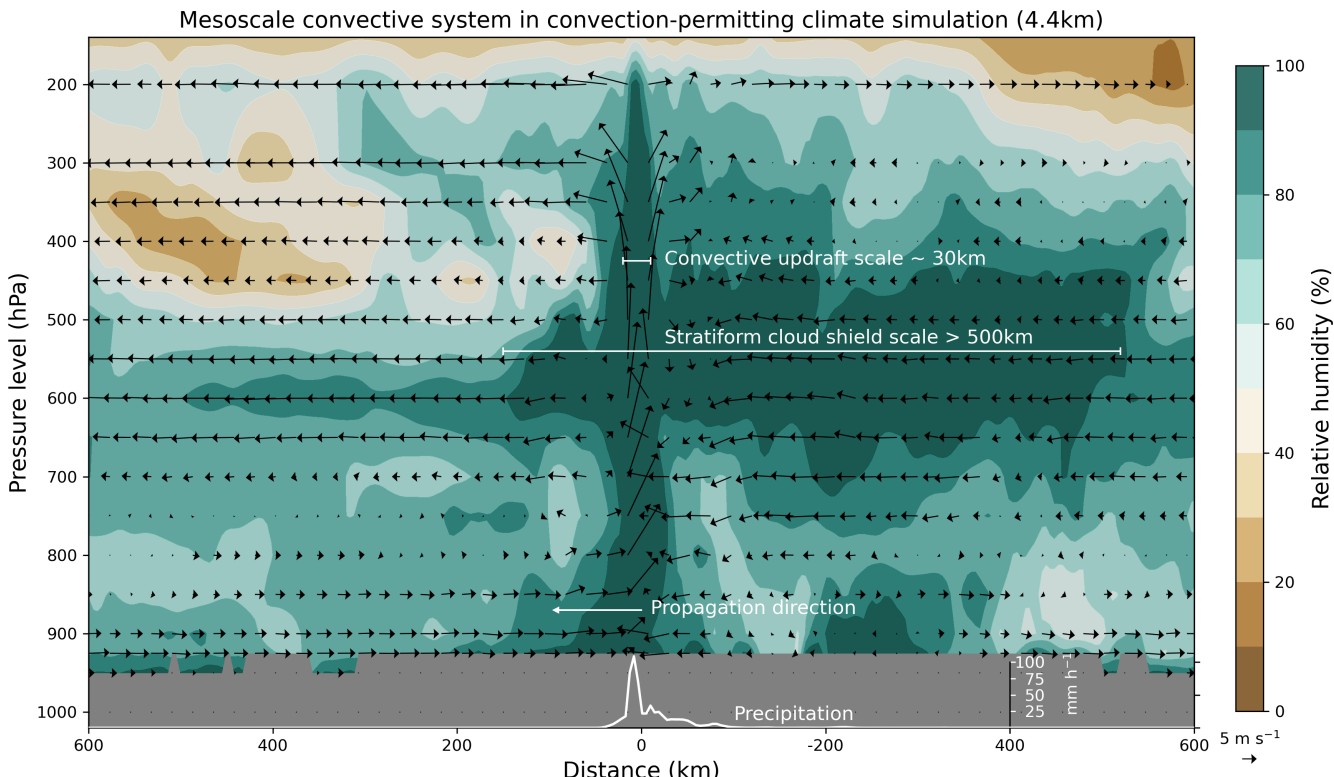

**Figure 2.** Explicit representation of convective storm circulations in a CP model. We show a simulated convective storm cross-section and for a single timestamp, centred on a storm updraft in a 4.4 km gridbox resolution convection-permitting climate model simulation (CP4-Africa, Senior et al., 2021). The organised storm is visible as an area of >90% relative humidity (shading), with extensive cloud anvil across 600-350 hPa pressure levels vertically and extending to a horizontal scale ('$x$'-axis) of > 500km. Wind vectors indicate a high vertical velocity at the cross-section centre point (0 km), extending across a horizontal scale of approximately 30 km, these being typical features resolvable in this high-resolution CP model but missing in climate models. The resolved updraft circulation is co-located with very high rainfall intensity locally at 0 km (white line, second bottom '$y$'-axis), which rapidly decreases as a function of distance and is specifically linked to the correct representation of the internal updraft circulation. These features of storm processes are expected to be sensitive to background global warming level (e.g. Prein et al., 2017).

Deriving equation sets via AI brings important challenges, first in the large number of potential input variables that influence convection. While temperature, humidity, wind, and pressure fields may serve as the baseline, derived quantities like Convective Available Potential Energy (CAPE) and other vertical profile descriptors, along with spatially variable land characteristics (e.g. topography, vegetation, land use, soil moisture) and oceanic features (e.g. sea surface temperature, surface roughness), can





significantly impact convective processes. Omitting these variables from any new equation set could hinder the transferability of knowledge from CP simulations. Moreover, equations must also accurately bridge the scale gap between CP and coarser models, necessitating an algorithm that discovers fundamental relationships transferable to ESM resolutions (e.g. Grundner et al., 2024).

Secondly, while equation discovery approaches offer promise in generating transferable equations for process descriptions beyond their training domain (Ross et al., 2023), it is important to verify they maintain physical consistency and adhere to fundamental principles like moisture and energy conservation. Thus, a question is whether both transferability and physical consistency challenges can be overcome by equation discovery targeting interpretability in ways that other methods cannot achieve. Expert judgement can constrain equation parameters within realistic physical limits, enhancing trustworthiness for

extrapolation beyond training conditions (Jebeile et al., 2023). The reward for deriving reliable equations for ESMs that capture convective behaviours is substantial. Reliable predictions of convective properties in future GHG-enriched environments are vital for policymakers to anticipate future rainfall extremes. Moreover, better constraints on upscale changes in circulation and radiative feedbacks linked to improved cloud cover modeling will lead to more reliable ESMs. A major concern is that some ESMs project very high simulated ECS values, however strongly depending on how they represent climate change feedbacks

on cloud features (Bjordal et al., 2020). Dufresne and Bony (2008) provide a detailed disaggregation of direct and feedback drivers (including changes to cloud characteristics) that contribute to simulated global warming as GHGs rise.

    Ultimately, as computing power advances, century-long global climate model ensembles at kilometre-scale may become feasible (Slingo et al., 2022), offering more robust projections of convection and related storms as GHG levels rise. However, given the urgent need to understand climate impacts at fine scales, an AI-supported approach is likely invaluable. Equation

discovery that captures local effects within a structure available for global calculations may offer an interim solution, reducing resource costs for large ensembles and uncertainty estimation, while providing crucial insights into future rainfall patterns.

### 3.2   Improving models of Terrestrial Carbon Cycling

We consider the task of modelling large-scale land-atmosphere carbon dioxide ($CO_2$) exchanges. A substantial fraction of $CO_2$ emissions are currently absorbed by the ocean and land surface, and their future extent affects global climate policy. Decreased

future natural "drawdown" implies that fewer emissions are compatible with any societal goal to restrict global warming to a target such as two degrees above preindustrial levels. However, the magnitude of these fluxes, even for the contemporary period is highly uncertain. This uncertainty is described in detail in many studies, including efforts to constrain it (e.g. Chandra et al., 2022). Budget calculations between emissions and atmospheric concentration changes reveal with high accuracy the combined global land plus ocean offset of emissions. However, Chandra et al. (2022) note (by citing Friedlingstein et al., 2020)

that the balance between the land and ocean components is unknown within the order of a GtC yr$^{-1}$. Approaches to reducing uncertainty in regional-to-global land-atmosphere $CO_2$ fluxes include using FLUXNET towers (e.g. Baldocchi et al., 2001) above strategic representative biomes, atmospheric $CO_2$ measurements merged with atmospheric transport models (generating atmospheric inversions, (e.g. Table 1 of Kondo et al., 2020)) and forward modelling with Dynamic Global Vegetation Models



(DGVMs) (e.g. Sitch et al., 2008). Robust forward modelling is of particular importance in quantifying flux changes expected for any altered climatic state.

The challenge of simulating the land surface is different from that of the atmosphere. One generalisation is that the equations and their parameters that describe atmospheric processes are well understood, but admit a particularly rich set of possible behaviours, including local convection, the effects of which are not understood at large scales (Section 3.1). The land surface, however, is modelled with simpler equations, including some components that are purely algebraic (i.e. not differential equations), but instead the complexity is substantial heterogeneity in their parameterisation. Variation in parameters can be due to multiple factors, including that a typical large-scale transect of land will contain many biomes or plant functional types, all having slightly different responses to imposed environmental variations. We propose AI-led approaches that quantify similar processes but with different magnitudes of response at finer scales. AI methods may also successfully aggregate spatial behaviours to generate equation parameters valid at much larger scales and thus amenable for inclusion in ESMs.

Eddy covariance is a measurement method that measures high-frequency (many times per second) simultaneous fluctuations in vertical windspeed and a scalar quantity of interest, and where the covariance statistic is linearly related to the land-atmosphere exchange of the scalar. In recent decades, there has been a growing number of towers with such measurement devices installed on top of them, estimating momentum, heat, vapour and $CO_2$ exchanges. The operation of eddy covariance systems over land and the related measurement databases are undertaken by the expanding FLUXNET network (Baldocchi et al., 2001). These measurements already provide training data for ML methods that map from global Earth Observation data products that record land attributes across to estimates of surface fluxes (Tramontana et al., 2016). This approach, named FLUXCOM, also entrains near-surface meteorological measurements as additional driving variables. FLUXCOM then extrapolates spatially, generating global historical estimates of surface energy fluxes (Jung et al., 2019) and $CO_2$ exchange (Jung et al., 2020).

Here, we suggest a slightly different approach to FLUXCOM. Using AI-led equation discovery without prior information will generate equations with strong similarities to existing knowledge, including established representations of surface energy partitioning (Monteith, 1981) and photosynthesis (Farquhar et al., 1980). However, two (or more) biomes are often in close proximity to each other, which has resulted in the development of "two source" models (e.g. Huntingford et al., 1995), or descriptions of biomes with complex canopy structures (e.g. Mercado et al., 2007). Here, we suggest an AI-led approach to building models of land-atmosphere $CO_2$ exchange, valid at the ESM gridbox scale and that account for extra local-scale complexities. We would first use equation discovery methods to model land behaviours for the footprint SCHMID (1994) of FLUXNET sites. Revised equations would map driving data from Earth Observation (EO) retrievals that fall within the flux tower footprint, along with FLUXNET meteorological measurements, to the tower data of land-atmosphere $CO_2$ exchange. The use of EO this way supports the suggestions of Chen et al. (2011). AI would derive equation terms and their parameters that are additional to current standard formulations (as might be in current ESMs) to capture surface heterogeneity factors. The dependencies on meteorological conditions support the generation of equations valid for altered background climatic conditions.





**(a)**

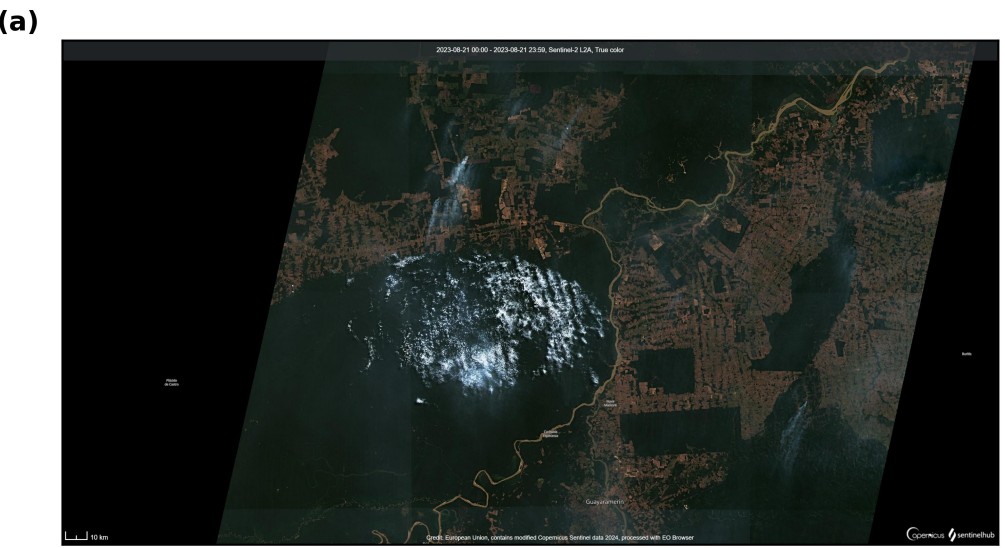

**(b)**

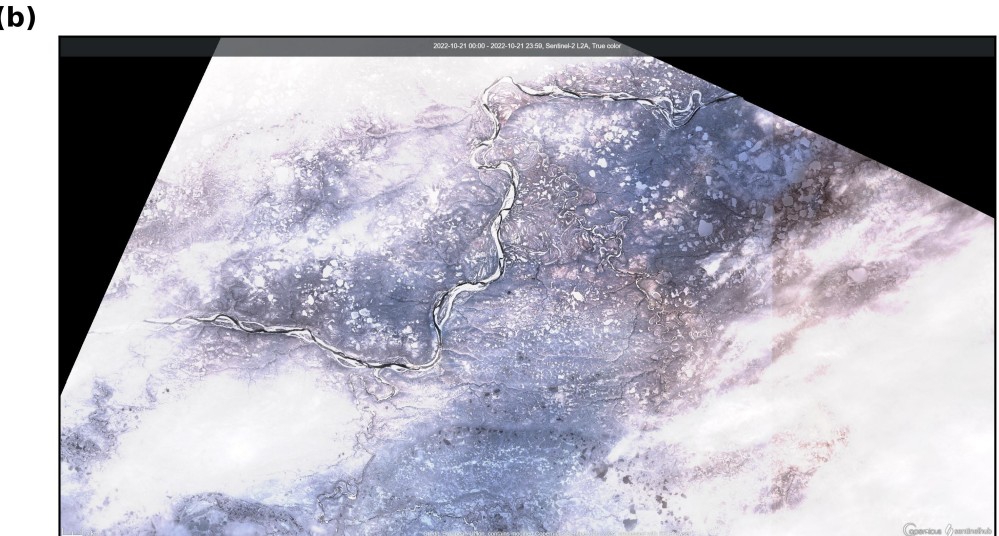

**Figure 3.** Sentinel-2 L2A images capturing complex land surface information down to 10m spatial resolution, in this example for (a) deforestation in Rôndonia, Brazil (10°S, 65.7°W) with visible wild fire plumes and (b) a permafrost landscape in Putorana State National Reserve, Russia (71°N, 96.3°E) (Sentinel-EO, 2024). A wealth of high-resolution imagery now documents processes acting on fine-scale land surface patterns and temporal changes therein, opening new avenues for AI-led mapping of sub-grid surface complexity onto physical variables typically used in climate models.





Once established at FLUXNET sites that AI-derived revised equations and their parameters successfully translate EO measurements to $CO_2$ exchanges, full images from EO data fields can be used as forcings elsewhere. That is, EO provides forcings to the equations derived at FLUXNET sites, to determine local fluxes away from towers, capitalising on the often complete spatial information held in satellite records. Deriving a careful spatial aggregation of these local calculations to the large ESM gridbox scale would provide equations and parameters in ESMs that capture more accurately fine spatial heterogeneity in the land surface, thus offering better predictions of $CO_2$ exchanges in such models. ML-derived spatial aggregation is a form of technique known as computer vision. In Figure 3 we present two representative images, panel (a) showing complexity in the South American tropical rainforest where there is extensive land use, and panel (b) of permafrost at high latitude, where there is substantial variation in land cover attributes. An additional requirement of computer vision algorithms is that they ignore locations in EO imagery where there are clouds or other masking factors such as smoke from fires (e.g. panel a of Fig. 3).

Our proposed approach would become increasingly accurate as the eddy covariance network extends, with Papale (2020) stating that FLUXNET expansion should occur to support an improvement in the accuracy of annual estimates of global land-atmosphere $CO_2$ exchange. Furthermore, the availability of EO data over ever-increasing time periods allows training (and more extensive testing) of AI-discovery equation approaches, including checking their performance at capturing climate-induced trends. Finally, as additional FLUXNET towers become live, it will be possible to more routinely test equations (rather than training algorithms to find them) at a range of locations. Where there are discrepancies, this may imply further missing processes in the equation set, or a strong regional dependency of parameters, which our techniques may help quantify. As an example, the introduction of geochemical cycles beyond carbon in land models is still in its early stages, with Davies-Barnard et al. (2022) noting major differences in its representation between ESMs.

## 3.3 Dynamical System Models of Ocean Circulation

The study of major oceanic circulations is conducted mainly with high-resolution numerical simulations, often as part of ESMs. However, the large computational time of such simulations maintains interest in faster summary models, mainly in the form of coupled Ordinary Differential Equations (ODEs). Reduced form operationally fast spatially aggregated bulk ODEs that evolve in time allow researchers to more readily scan parameters and a broader range of future climate forcings, enabling a better assessment of potential features of circulation stability. These simpler dynamical systems can enable levels of understanding not possible with restricted computer power constraining the number of possible perturbed parameter full complexity simulations.

Early model attempts at simplified descriptions of oceanic behaviour exist including the Atlantic Meridional Overturning Circulation (AMOC) (e.g. Stommel, 1961). More recently, simplified models have emerged that include atmospheric drivers and their impact on the important El-Niño-Southern Oscillation (ENSO) (e.g. Timmermann et al., 2003). Here, we present the Timmermann model in Fig. 4, in both schematic form (panel a), and bifurcation diagram (panel b), with details in the caption.

ENSO is hypothesised to occur as follows. There is a positive ocean-atmosphere feedback process that activates ENSO, first suggested by Bjerknes (1969). The feedback process may start with weakened easterly trade winds, which reduces the strength of the ocean current responsible for drawing surface water away from the western equatorial Pacific. This in turn reduces the ocean up-welling of colder water from the deep ocean, flattening the thermocline. A build up of warmer surface water in





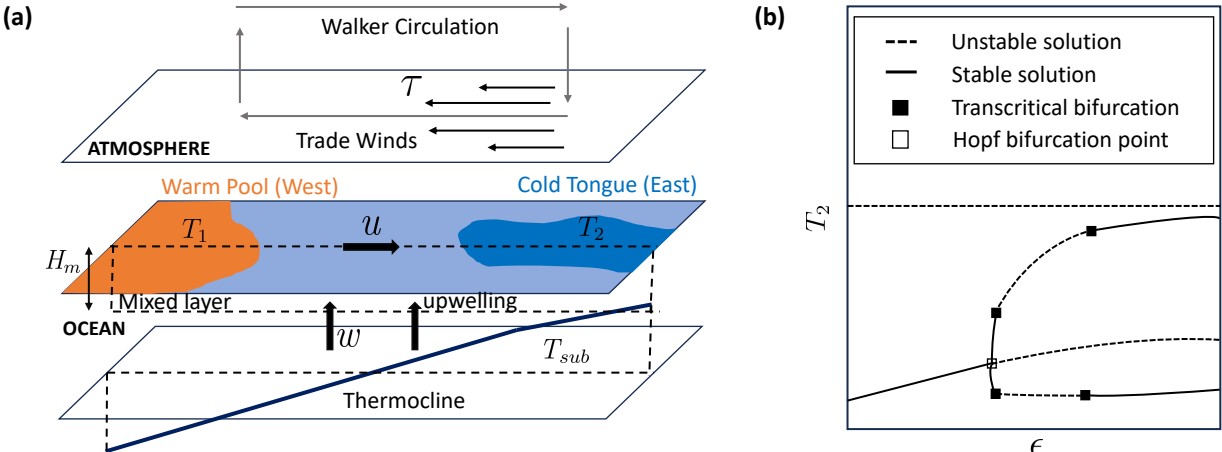

**Figure 4.** A schematic and bifurcation diagram of the equatorial coupled ocean-atmosphere system as represented in the Timmermann model of ENSO. Panel (a): $T_1$ (K) and $T_2$ (K) are the sea surface temperatures of the western Pacific and eastern Pacific respectively, and $\tau$ is the wind stress on the ocean surface due to easterly trade winds, given in Newtons per meter squared. $T_{\mathrm{sub}}$ (K) is the temperature below the mixed layer of depth $H_{\mathrm{m}}$ (m). The ocean upwelling velocity is denoted $w$ (m s$^{-1}$) and $u$ (m s$^{-1}$) is the atmospheric zonal surface wind. Diagram adapted from Dijkstra (2013). Panel (b): Bifurcation diagram of eastern Pacific temperature $T_2$ as a function of zonal advection efficiency $\epsilon$, showing solutions to Eqs. (3) to (4) and their stability. Diagram adapted from Timmermann et al. (2003).

the equatorial east Pacific (El Niño) then emerges. As a result, we now have a reduced east-west SST gradient that further weakens the Walker circulation (a positive feedback mechanism). However, after the El Niño matures, a negative feedback mechanism emerges to turn El Niño into a cold phase known as La Niña. This negative feedback mechanism accounts for the observed oscillatory behaviour of the coupled ocean-atmosphere ENSO system which has a characteristic timescale of two

5    to seven years. In addition, the tropical Pacific sea surface temperature (SST) also exhibits decadal variability (Timmermann et al., 2001). Previous studies have shown this pattern has two leading modes of inter-decadal variability, the ENSO-like and ENSO-induced modes (Choi et al., 2012), where the latter mode is strongly related to decadal variations in the amplitude of ENSO.

       When modelling the processes that cause irregular inter-annual El Niño occurrences, there are usually two approaches. The

10   first is deterministic, albeit that there exists chaotic behaviour in the large-scale dynamics of the coupled ocean-atmosphere system due to nonlinear interactions. The other viewpoint assumes that this behaviour is only weakly nonlinear and the irregular but oscillatory feature is mainly due to stochastic noise. The former approach is more suited to using reduced complexity models of ENSO, which may be gained from using AI-led equation discovery methods to uncover chaotic nonlinear dynamics without the need for stochastic terms in the equations. We now describe some of the deterministic modelling of ENSO that

15   exists already.

       Modelling the positive and negative feedback mechanisms of ENSO initially led to four basic deterministic oscillator models. Such models are known as the delayed oscillator (Suarez and Schopf, 1998), the recharge–discharge oscillator (Jin, 1997a),





the western-Pacific oscillator (Weisberg and Wang, 1997) and the advective–reflective oscillator (Picaut et al., 1997). All four of these models are linear, producing periodic oscillations. However, observed ENSO behaviour has been shown to exhibit nonlinear and irregular oscillations, which can only be modelled with these four linear models if external noise forcing is applied. As a result of observing this irregular behaviour, nonlinear deterministic models of ENSO have also been built to

capture the chaotic behaviour and without the need for an annual noise forcing to generate it. One such model is that of Timmermann et al. (2003), which is a dynamical systems model that captures both the inter-annual oscillations and the decadal variability of El Niño events seen in observations and climate models. The physical setup involves using a two-strip and two-box approximation (Jin, 1997b), extending the simpler ENSO description proposed by Zebiak and Cane (1987). The upper equatorial ocean is described using a box model version of a shallow-water model in conjunction with a mixed ocean layer of

fixed depth. The heat budget is given by two coupled first order ordinary differential equations in time, as

$$\frac{dT_1}{dt} = -\alpha(T_1 - T_r) - \frac{u(T_2 - T_1)}{L/2} \tag{3}$$

$$\frac{dT_2}{dt} = -\alpha(T_2 - T_r) - \frac{w(T_2 - T_{\text{sub}})}{H_m} \tag{4}$$

where the variables $T_1$, $T_2$, $T_{\text{sub}}$, $H_m$, $w$ and $u$ are described in the caption to Fig. 4. The additional variables in Eqs. (3) and

(4) are as follows. $T_r$ (K) is the radiation equilibrium temperature in kelvin, $L$ (m) is the basin width and $1/\alpha$ (day$^{-1}$) represents a typical inverse timescale of thermal damping. Furthermore, we have the physically derived relations $\frac{u}{L/2} = \epsilon\beta\tau$ and, $w/H_m = -\zeta\beta\tau$ where $\epsilon$ and $\zeta$ are the strengths of zonal and vertical advection respectively (model bifurcation parameters).

The subsurface temperature, $T_{\text{sub}}$ is defined as,

$$T_{\text{sub}} = T_r - \frac{T_r - T_{r0}}{2}\left[1 - \frac{\tanh(H + h_2 - z_0)}{h_*}\right] \tag{5}$$

where $h_2$ is the east equatorial Pacific thermocline depth (relative to a depth $H$) in meters, $z_0$ is the depth in meters, for which the upwelling velocity $w$, in meters per second, becomes its characteristic value and $h_*$ is the sharpness of the thermocline. The thermocline depths are calculated as follows:

$$h_2 = h_1 + bL\tau \tag{6}$$

$$\frac{dh_1}{dt} = -r\left(h_1 + \frac{bL\tau}{2}\right) \tag{7}$$





where $h_1$ is the west equatorial Pacific thermocline depth in meters, $b$ is the efficiency of wind stress $\tau$ to drive the thermocline tilt. Wind stress is given by

$$\tau = \frac{\mu(T_2 - T_1)}{\beta}. \tag{8}$$

Overall this leaves us with six equations (Eqs. (3) to (8)) and six unknown variables ($T_1$, $T_2$, $T_{\text{sub}}$, $h_1$, $h_2$ and $\tau$). The
original parameter values used in the study Timmermann et al. (2003) were $T_{\text{r0}} = 16\,°\text{C}$, $T_{\text{r}} = 29.5\,°\text{C}$, $\alpha = 1/180\,\text{day}^{-1}$, $r = 1/400\,\text{day}^{-1}$, $H_{\text{m}} = 50\,\text{m}$, $H = 100\,\text{m}$, $z_0 = 75\,\text{m}$, $h^* = 62\,\text{m}$, $\mu = 0.0026\,\text{K}^{-1}\,\text{day}^{-1}$, $\mu bL/\beta = 22\,\text{m}\,\text{K}^{-1}$, $L = 15 \times 10^6$ m and $\zeta = 1.3$. In Timmermann et al. (2003), the bifurcation parameter, $\epsilon$, is varied to understand how the nonlinear zonal advection term generates chaotic behaviour. The typical values of $\epsilon$ range between 0.024 to 0.24, as calculated from CGCM simulations and ocean data assimilation products (Timmermann et al., 2003).

This low-order model simulates strong decadal El Niño conditions (El Niño bursting) which have been observed in full complexity simulations, where the ENSO mode grows in amplitude and then quickly resets, from which the amplitude variations grow again. For small values of the zonal advection efficiency bifurcation parameter, $\epsilon$, and a fixed value of $\zeta$, the system is in stable equilibrium with a cold "tongue" in the eastern Pacific and a warm pool in the western Pacific, as shown in Fig. 4 Panel a). However, this steady mean state becomes unstable for a larger critical value of $\epsilon$, leading to a Hopf bifurcation, and a stable
periodic orbit appears, as illustrated in Fig. 4 Panel b). For simplicity we have described here the situation of varying $\epsilon$ and keeping $\zeta$ constant. Further analysis (Fig. 3 in Timmermann et al. (2003)) finds similar behaviour for $T_2$ as a function of $\zeta$ also giving rise to stationary and oscillatory solutions of Eqs. (3) and (4). Key to this is for values of $\zeta > 0.54$ a Hopf bifurcation emerges giving rise to the oscillatory self-sustained ENSO mode with a typical period of several years.

The full three-dimensional ENSO system exhibits both periodic and chaotic windows during periods of ENSO amplitude
modulations, for instance when the efficiency of zonal advection, $\epsilon$, takes on larger values. Further rich dynamical behaviour exists when both $\epsilon$ and $\zeta$ are varied simultaneously, also giving rise to ENSO amplitude modulations, period-doubling bifurcations and chaos (Timmermann et al., 2003). Relating to the premise of this paper, although the existing simpler models such as that repeated above appear to reproduce many features of ENSO, utilising AI-led equation discovery offers a potential way to verify this dynamical behaviour using observational data sets and model simulations as training data. Do these newer
algorithms back out a model with implicit Hopf bifurcations and low-dimensional chaos that match that found in Timmermann et al. (2003), or will they suggest refinements?

We propose the three main potential outcomes of using equation discovery in this context. Firstly, equation discovery may back out simpler deterministic models of ENSO, such as the four basic linear oscillator models mentioned previously. Most of these models can be generalized into a recharge oscillator framework (Jin and An, 1999). Secondly, with careful consideration
of the variables and the physical set-up involved, included in Fig. 4, the derived equations may provide verification of the Timmermann model by exhibiting similar nonlinear chaotic behaviour for different values of the bifurcation parameters. If this is the case we expect the discovered equations to emulate 1) low-amplitude biannual ENSO oscillations at low values of $\epsilon$, 2) amplitude-modulated and chaotic behaviour at intermediate values of $\epsilon$ and 3) large amplitude ENSO oscillations with periods





of 3-4 years at higher values of $\epsilon$. A third possibility is that the learned equations provide a new description of ENSO behaviour which may potentially come with 1) better parameterisation of the ENSO system, 2) enhanced understanding of physical mechanisms leading to the generation of nonlinear ENSO behaviours which agree better with observations and models and, 3) improved predictive skill of ENSO.

Due to the shortness of available observational records and the level of noise in the data, it can be difficult to determine from observations alone if ENSO amplitude modulations are a statistical manifestation or a result of deterministic processes (Timmermann et al., 2003). AI-led equation discovery methods such as sparse regression only require a limited time series to discover the underlying dynamics and work well in the presence of noise, and therefore may be effective in this situation.

       State-of-the-art ESMs still contain biases in the eastern equatorial Pacific (Timmermann, 2018) which leads to problems in
representing key physical processes, feedback mechanisms and so may impact their ability to accurately predict future changes. ESMs struggle to converge in their simulations of ENSO characteristics such as amplitude, period and the asymmetry between El Niño and El Niña phases (Jin et al., 2020). Yet, inter-ESM disagreements are an opportunity for AI-led discovery of reduced complexity equations, as such models are computationally fast and thus allow substantial sampling of different parameter values. If such parameters have process interpretations and if it is possible to map ESM projections onto different values of
such parameters, this enables the differences and uncertainties between ESMs to be better understood.

## 4    Discussion and Conclusions

AI plays an increasing role in society (e.g. Makridakis, 2017) and will likely influence multiple aspects of scientific research (e.g. Xu et al., 2021). Artificial intelligence methods are proposed to support the mitigation of climate change (e.g. Kaack et al., 2022; Chen et al., 2023; Rolnick et al., 2023), and climate research itself (e.g. Reichstein et al., 2019; Huntingford et al., 2019;
Eyring et al., 2024). The basis for most AI algorithms is statistical, frequently introducing nonlinearity into forms of regression (e.g. Murphy (2013), with an early chapter on linear regression, to be viewed in the context of subsequent chapters on key AI algorithms). Recently, there has been an emphasis on constraining AI-based discoveries to be compatible with known theory about underlying processes. These methods are referred to as "physics-informed AI" (e.g. Karniadakis et al., 2021) and are considered for climate analysis (e.g. Karniadakis et al., 2021). An early form of physics-informed activity has been the de-
velopment and maintenance of data assimilation methods to generate historical gridded datasets of meteorological conditions, e.g. the European Centre for Medium-Range Weather Forecasts ERA5 reanalysis (Hersbach et al., 2020). Reanalysis products merge data with forecasts, balancing the need to simultaneously remain within the uncertainty bounds of measurements and broadly satisfy the equations of atmospheric dynamics.

       We propose advancing climate change science with the even newer AI technique of "equation discovery". Although existing
AI methods provide powerful insights into the features of large datasets, they remain embedded in statistical approaches. Unfortunately, statistical descriptions may not estimate well "out-of-sample", yet the main requirement of climate science is to predict environmental regimes different from those of the present day or the recent past. If, instead, AI reveals process equations, this opens the opportunity for their assessment and parameterisation. This understanding of processes may confirm





that equations have the predictive capability needed to describe new climatic regimes. Indeed, a fundamental role of applied mathematics is to determine data-led equations for a system that can subsequently simulate responses to alternative forcings.

Already suggested is using AI to emulate the few limited-area very high resolution atmosphere simulations, and for the forcings for which they have been operated (e.g. Schneider et al., 2023). Such emulators may be placed in ESMs to predict
high-resolution atmospheric features, including average storm characteristics, at other locations. Extending such analyses to generate governing equations (possibly with stochastic components) may more rigorously capture high-resolution effects at alternative places and atmospheric GHG levels. Equation sets are amenable to discretisation in ESMs and including feedbacks. We also consider extending basic equations, allowing additional perturbation term discovery, or location-specific parameters, e.g. for capturing ecological responses with strong spatial heterogeneity. A further application is AI-based equation discovery
valid over substantial areas (e.g. averaging ESM projections), aggregating geographical variation and reducing to Ordinary Differential Equations (ODEs) in time only that respond to changing GHGs. ODEs are often amenable to more complete analysis, and even if exact solutions are unavailable, they can be tested for linear stability about equilibrium states, and for nonlinear systems including how parameter perturbation may activate tipping points. TPs in the Earth system are the basis for intense research (e.g. Mckay et al., 2022), given their strong potential impacts on society, yet estimated GHG levels causing
occurrence are highly dependent on ESM studied (Drijfhout et al., 2015). Mapping to a common simpler model enables characterising ESMs by effective parameters in such reduced complexity representations. This approach may identify parts of the Earth system that, if measured better, will better determine GHG levels likely to trigger TPs.

Could AI replace conventional climate research? This question is already being asked about weather forecasting (e.g. Schultz et al., 2021). AI is shown to have good skill in predicting the emergence of severe weather events (e.g. Bi et al., 2023; Lam et al.,
2023), but McGovern et al. (2017) argue that the use of such methods to assess high-impact meteorological occurrences should be performed in parallel with physical understanding. A deeper understanding of the balance of dominant equation terms, possibly determined by AI, may reveal causal links between processes ("storylines"; Shepherd (2019)) during the preceding periods of extreme events and thus provide earlier warnings of their occurrence. If AI methods extend to the extra step of deriving underlying equations, this offers potentially new insights, but still requires substantial human interpretation to achieve
them. Equation discovery will engage climate research scientists further, rather than creating any form of replacement.

In summary, in Section 3 we offer three illustrative examples of how AI-led "equation discovery" may support climate change science, as:

– **Simulation of atmospheric convection**. Convection modelling involves numerically solving equations at very high resolution to accurately represent the spatial heterogeneity of individual storms. Typically, a grid spacing of about one
30       kilometre is necessary, leading to high computational demands. Consequently, simulations are limited in both spatial extent and time period simulated, the latter restricting the range of GHG concentrations modelled. The goal is to develop simplified "bulk" differential equations suitable for ESMs, hence using coarser grid spacings (around 100 kilometres) that accurately aggregate fine-scale dynamics and their interactions with boundary conditions. Ideally, the equations also feed back, where appropriate, on ESM-simulated large-scale dynamics. Incorporating these equations into ESMs enables
35       global convection representation under broad ranges of atmospheric GHG levels and at different locations. The equations





may incorporate statistical or stochastic components to describe the intensity and duration of convective events at fine scales or at single points.

- **Simulation of terrestrial carbon cycling**. In this case, the general governing equations for land-atmosphere $CO_2$ exchange due to photosynthesis and respiration are known, broadly valid at large spatial scales and routinely included in ESMs. However, at very local scales, the parameterisation of these equations depends strongly on biome type. There may also be a need to derive additional equation terms for complex canopy structures or where different biomes are so close that there are key within-canopy interactions. Hence, the challenge is two-fold. First, to calibrate and, where appropriate, expand equation terms for key biomes or colocated biomes, possibly guided by eddy-covariance measurements. The second is to again use AI methods to entrain Earth Observation data, enabling spatial aggregation away from flux towers to generate equations and parameters applicable at large ESM gridbox scales. This enables ESMs with fully interactive carbon cycle simulation to better assess the extent the surface of the land will partially offset future $CO_2$ emissions.

- **Very large-scale summary simulation of ocean currents**. This AI application aims to derive globally applicable reduced-complexity models from ESMs that simulate oceanic currents at a smaller scale, offering several benefits. A simpler model can explore wide ranges of forcings, such as changes caused by many different future emissions trajectories and that cannot be simulated with full ESMs due to their computational constraints. Simpler models facilitate parameter scanning and, in the case of nonlinearity, enable broad-scale representation of how TPs may occur. Fitting simpler model parameters to the individual members of ESM ensembles enables better characterisation of differences and uncertainties between ESMs. New AI-discovered reduced complexity equations from data or ESMs might indicate whether existing simpler models, such as that of Timmermann describing ENSO, remain suitable or whether alternative variants of oscillator models are more appropriate, at least for some ESM frameworks.

It is relatively easy to set aspirations for implementing AI methods in climate science, rather than performing the analysis itself. Some suggestions here are likely major research projects that could take multiple years to complete. However, with the rapid pace of algorithm development raising questions about applicability to climate research, we highlight the particular method of equation discovery. We contend that equation discovery, a form of interpretable AI, can substantially enhance climate research in ways not possible by traditional analytical or statistical methods. An emphasis on equation development and their inherent description of processes moves on from the complaint that AI-developed models are purely statistical and may fail if extrapolated beyond current forcings to higher GHG levels. Although an initial equation set may also be suspected to have poor "out-of-sample" capability, its existence provides a stronger basis for understanding processes and interactions. Subsequent more careful fitting of equation parameters may generate predictive capability.

ESMs will undoubtedly continue as the main tool for advising climate policy. Two of our examples (convective storm modelling and terrestrial carbon cycling) offer the possibility to enhance ESM reliability. Better aggregation of subgrid storm process representation to gridbox scale may remove known issues with existing cloud representation (Randall et al., 2003). The current spread in subgrid atmosphere and land parameterisations may be contributing to the large inter-ESM differences in the projection of changes in rainfall patterns (e.g. Yazdandoost et al., 2021) and the global carbon cycle (e.g. Huntingford





et al., 2009) respectively. Improving emulation of subgrid effects to support ESM development fits with the commentary of Wong (2024) on AI and climate, although we stress retaining process understanding through equation representation.

We believe that AI does not represent a threat to climate science and will instead support researchers to increase their skill to predict meteorological conditions as atmospheric GHGs rise. Climate change is simulated with ESMs, yet their ability to
offer ever more refined estimates of change is arguably at a standstill. For similar future pathways in GHGs, the spread of projections between the models in version 5 of the Climate Model Intercomparison Project (CMIP5) and the more recent version 6 (CMIP6) has not decreased for basic quantities of changes in global mean temperature and global mean precipitation (e.g. Fig 4 of Tebaldi et al., 2021). Reflecting our three examples (Sections 3.1, 3.2 and 3.3), we conjecture that the particular form of AI which is the discovery of equations may reduce these uncertainties. Reductions will be by (i) providing new robust
equations that capture subgrid processes, (ii) creating valid grid-scale parameterisations for existing equations that aggregate fine-scale processes and (iii) disentangling complex processes to equation sets far simpler than ESMs but that capture the dominant processes. The latter simpler equations may guide measurement programmes towards tuning key parameters, and where such knowledge ultimately feeds back by improving ESM parameter calibration.

## 5  Code availability

This is a review article and therefore does not contain computational elements.

## 6  Data availability

No new data are used in this manuscript. The manuscript contains modified Copernicus Sentinel data (2024) processed by Sentinel Hub: https://apps.sentinel-hub.com/eo-browser/.

*Author contributions.* CH devised the original concept and framework and selected the examples for the manuscript. AJN wrote the section
on simplified oceanic modelling, CK wrote the section on aggregating atmospheric convective events, and JAA supported the description of existing ML/AI algorithms. All authors contributed to the general writing of the manuscript.

*Competing interests.* The authors confirm they have no competing interests.

*Acknowledgements.* C.H. acknowledges the Natural Environment Research Council (NERC) National Capability award to the U.K. Centre for Ecology and Hydrology. A.J.N. is grateful for NERC funding through the Oxford University Environmental Research Doctoral Training
Partnership (DTP). C.K. acknowledges funding from a NERC independent research fellowship (NE/X017419/1).





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
