# Peer review of "Potential for Equation Discovery with AI in the Climate Sciences"

_Earth System Dynamics, 2024_

## Referee Comment (RC1)

**Review**

Title: Potential for Equation Discovery with AI in the Climate Sciences
Author(s): Chris Huntingford, Andrew J. Nicoll, Cornelia Klein, and Jawairia A. Ahmad
MS No.: esd-2024-30
MS type: Perspective
Iteration: Minor revision

Verdict: Minor Revisions

**SUMMARY:**
This interesting paper introduces the promising research field of AI-led equation discovery to climate science. The authors provide an extensive overview over previous and current statical methods including machine learning based approaches in the field of climate science. As a remedy to the current issues such as transparency of most fully data-driven approaches and computational limitations of physics-based numerical solutions, the authors suggest the application of "equation discovery" algorithms. These AI-based algorithm enable equation generation for unknown dynamical system as well as for systems with limited dynamical information. Focussing on symbolic regression methods and specifically the SINDy algorithm, the authors provide a comprehensive, understandable and detailed description of the procedure of equation discovery. Lastly, the examples of potential applications, such as in atmospheric convection, carbon cycle parameters, and ocean feature modelling for assessing tipping point risks, further strengthen the author's conclusion and outline promising research avenues.

**RESPONSE:**
Overall, I find this submission to be a relevant contribution to the climate science community as it provides a comprehensive overview of ongoing research, outlines current road-blocks and specifically suggests promising approaches which were previously over-looked or unknown in the field. Therefore, I highly suggest to improve upon the clarity and structure of the abstract, introduction and conclusion. This is necessary, as I find the importance of this contribution sometimes gets lost in overly long and disjunct paragraphs and sentences. While, I like the designated re-iteration of the potential application examples in the conclusions, I suggest to improve these by focussing on the discussed ways of using equation discovery in each application. I address my concerns in details below and discuss further individual remarks.

1. *Relevance* (Does the paper address relevant scientific questions within the scope of ESD?)
This manuscript puts forward a new avenue of potential machine learning based earth system model (EMS) research, by suggesting AI-based equation discovery. Based on three highly relevant potential application examples, the authors demonstrate the relevance and motivate the integration of the proposed research direction.

2. *Novelty* (Does the paper present novel concepts, ideas, tools, or data?)
While the idea of equation discovery is an established sub-domain of machine learning and the discussed methods long-standing, their application to earth system modelling is a novel idea, to the best of my knowledge.

3. *Substantial conclusions* (Are substantial conclusions reached?)
The authors clearly establish their conclusions regarding the capabilities and potential of the AI-led equation discovery, throughout the manuscript

4. *Clarity and Validity* (Are the scientific methods and assumptions valid and clearly outlined?)
While the manuscript does exhibit structural and literary weaknesses, the algorithms in section 2.2 as well as potential applications in Section 3.3. are relayed in details and in an understandable manner.

5. *Support of the interpretations and conclusions* (Are the results sufficient to support the interpretations and conclusions?)
The presented potential applications sufficiently support and highlight the relevance of the presented methods for climate science and earth system modelling.

6. *Traceability of results* (Is the description of experiments and calculations sufficiently complete and precise to allow their reproduction by fellow scientists ?)
As this paper aims to be more of a perspectives/review paper the authors do not present new experiments and therefore do not require exact reproducibility.

7. *Consistency of related work* (Do the authors give proper credit to related work and clearly indicate their own new/original contribution?)

The authors consistently provide necessary citations and clearly establish the reviewed material as well as their own contributions.

8. *Title* (Does the title clearly reflect the contents of the paper?)

The manuscript title fully aligns with the manuscript content.

9. *Abstract* (Does the abstract provide a concise and complete summary?)

The abstract provides a full summary of the manuscript content. However, I highly recommend to improve structure and overall writing, as it is hard to read and difficult to follow, which does not reflect the relevance and value of this submission.

10. *Structure and Clarity* (Is the overall presentation well structured and clear?)

While I enjoyed the explanatory figures and graphs, especially introduction and discussion should be improved to further strengthen the value of this manuscript. Overly long sentences alongside sometimes disjunct paragraphs and sentences make these sections hard to follow. Exceptions are Section 2.2, 3.1, and 3.2 which, while long were easy to read.

11. *Language* (Is the language fluent and precise?)

I do have some concerns regarding the clarity of the paper (see previous point).

12. *Math* (Are mathematical formulae, symbols, abbreviations, and units correctly defined and used?)

In Section 2.2 and Section 3.3, most mathematical expression are well-defined and explained. I only have minor concerns, which I detailed below.

13. *Possible Reduction* (Should any parts of the paper (text, formulae, figures, tables) be clarified, reduced, combined, or eliminated?)

The manuscript would profit from a more precise and reduced section 2.1. (Background), discussion and introduction. In addition these parts should also be rewritten to improve clarity and readability, which currently hampers the value of this interesting contribution.

14. *Number and quality of references* (Are the number and quality of references appropriate?)

The authors clearly cite all relevant works and choose relevant works out of the respective fields. However, while SINDy and symbolic regression is a well-renowned, the field of equation discovery also extends to more novel and promising algorithms, e.g., neural operators (Lu et al. 2021, Cao et al. 2023)

15. *Supplement* (Is the amount and quality of supplementary material appropriate?)

I find this paper to be fully self-contained and therefore see no need for supplementary material.

**MINOR CONCERNS:**
1. *Disjunct sentences*: I find some sentences to be very hard to read, e.g. (p. 21 l. 8-10) "First, to calibrate…"

**TECHNICAL CORRECTIONS:**
1. Sec. 2.2: Please clarify the dimensionality of $\xi$ (l. 6 p.8) is it the same as $\xi\_2$. In addition I think a further specification of y might be helpful (l.7 p.8), since apparently $y = \Theta\xi$?
2. Sec. 3.3: Please add definition/descriptions of $\beta$ and Tr0, since they appear to not be defined.
3. L. 13-15, p. 12: "A major concern…" -> This sentence is not understandable please check the sentence structure.

Cao, Qianying, Somdatta Goswami, and George Em Karniadakis. "LNO: Laplace neural operator for solving differential equations." *arXiv preprint arXiv:2303.10528* (2023).

Lu, Lu, et al. "Learning nonlinear operators via DeepONet based on the universal approximation theorem of operators." *Nature machine intelligence* 3.3 (2021): 218-229.

---

## Community Comment (CC1)

The paper "**Potential for Equation Discovery with AI in the Climate Sciences**" is a vital discussion topic for advancing climate research. It's clear that there are infinitely many more non-linear formulations than the linear set of possibilities that humans are comfortable with solving. Fluid dynamics *a la* Navier-Stokes by itself contains many non-linear elements that have not been completely explored due to a lack of ability to solve in a closed form.  The paper suggests an important possible constraint to apply: *"For physical systems involving fluid flows where the underlying equations are known to be energy-preserving, although also nonlinear".*

And that's where artificial neural networks and symbolic regression (i.e. equation discovery) comes into play. There are really few other alternatives outside of tedious human trial & error that are available to both (1) fully explore the combinatorial solution space and (2) incorporate numerical solvers to train the possible solutions to fit the available data using appropriate metrics for plausibility and precision.

The paper as is falls short on two fronts, one of which the authors' themselves highlight.  The first can be remedied by citing the importance of *cross-validation* (CV) strategies.  The success of machine learning is in part due to how CV can separate the wheat from the chaff in potential solutions. Yet, nowhere in the text is cross-validation mentioned, and this is a vital part of equation discovery, as an optimal CV algorithm+metric is necessary to isolate candidate solutions along a Pareto front of complexity (1/plausibility)  vs precision.  Neural networks can fit just about any curve, so CV approaches to equation discovery help to eliminate those that are the result of over-fitting. Suggest Ref [1] as a citation starting point.

The second front is based on the authors' statement *"It is relatively easy to set aspirations for implementing AI methods in climate science, rather than performing the analysis itself"*.  I read this as a call to just do it instead of dreaming it, or as the thespian philosopher Christopher Walken said: "If you want to learn how to **build a house**, then **build a house**. Don't ask anybody. Just **build a house**."  The paper suggested
*"We discuss the potential application of AI-led equation discovery to three Earth system components. In each example, there is presently a deficiency in understanding, causing uncertainty in the representation of processes by equations. Each application falls into one of three categories. "*

Instead, I would recommend three Earth system components to evaluate: solid body, atmosphere (gas fluid), and ocean (liquid fluid).  In our text Mathematical Geoenergy, P. Pukite, D. Coyne, D. Challou (Wiley/AGU, 2019),  we describe novel equation-based models  for the Earth's Chandler wobble (solid body), QBO (atmosphere),and ENSO (ocean). The original nonlinear models were derived from simplifying Euler equations of motion for the Chandler wobble, and Laplace's Tidal Equations, which are simplified Navier-Stokes, for QBO and ENSO. We attain excellent agreement against observations in each case, and this extends to other climate indices such as AMO and PDO. See Figures 1..X at the end of this review.

Over the past few years, I have tried various machine learning approaches including neural networks and symbolic regression to observe if they would "discover" the same equation solutions I had formulated and applied.  First, it's clear that neural networks can't do the job as they train only on their own data-set as supplied, and so won't automatically pull in all the tidal time-series data available. This is the *closed-world assumption* (CWA) problem well-known in AI circles for years, see Ref [2]. Neural networks will fit the data, but it's all based on dreaming up patterns from the data instead of tracing it

back to a non-linear modulation from an external forcing. Alas, that external data set doesn't exist in the training data, so it gets ignored.

The symbolic regression/equation discovery approaches do an arguably better job. Although they also suffer from the CWA problem, they can make up for it by creating symbolic expressions from their library of primitive mathematical operators to draw from, such as creating a tidal forcing from (1) the time base, (2) arbitrary constants, and (3) sinusoidal primitives such as sin() and cos(). So, in terms of results, the frequencies from tidal factors do emerge in a symbolic regression fit to QBO, yet they are not directly harmonically-related due to the intrinsic non-linearity of the equation solutions! Thus, they may easily get overlooked when the symbolic regression results are deconstructed, as it also requires knowledge of nonlinear signal processing concepts such as aliasing and side-banding. That's what I have found straightforwardly in the Chandler wobble and QBO results, and with more of a challenge in the oceanic indices such as ENSO. The symbolic regression tools that I have evaluated include Eureqa, PySR, and TuringBot, Ref [3].

And this reflects back on the importance of cross-validation approaches and the selection of correlation metrics, including those that have proved valuable in machine learning in the context of noise and uncertainty, such as dynamic time warping - Ref [4] and complexity-invariance distance - Ref [5]. The results of symbolic regression depend on the best metric for the data, as some may prove too stiff to emerge from a local optima.

I agree with the paper that the focus on statistical machine learning to model climate variation is misguided, as it is more evident that large scale behaviors that are the result of collective deterministic actions describe better the standing wave models of ENSO and QBO. These will show the detail and variety in waveforms captured by wave equations, not the smeared responses captured by statistical ensembles.

Moreover (and finally), it is difficult to get a new paradigm accepted in geophysics fields such as climate science unless the results are beyond reproach. The complete lack of controlled experiments to test novel equation-based models means that claims of excellent agreement are dealt with suspicion. It is costly in terms of money and time to wait years for predictive models to come true, so the hope is that cross-validation results can conclusively demonstrate a new equation formulation has merit.

**Ref**

[1] Sweet, L., C. Müller, M. Anand, and J. Zscheischler, 2023: Cross-Validation Strategy Impacts the Performance and Interpretation of Machine Learning Models. *Artif. Intell. Earth Syst.*, **2**, e230026, https://doi.org/10.1175/AIES-D-23-0026.1.
[2] Zhu, Fei, Shijie Ma, Zhen Cheng, Xu-Yao Zhang, Zhaoxiang Zhang, and Cheng-Lin Liu. "Open-world machine learning: A review and new outlooks." *arXiv preprint arXiv:2403.01759* (2024).
[3] In my opinion, Eureqa did the best job but it has not been available for use since 2017 as it was sold to an AI firm for proprietary use. https://en.wikipedia.org/wiki/Eureqa , https://turingbotsoftware.com/ which is a attempted clone of Eureqa. https://github.com/MilesCranmer/PySR
[4] Li, Hailin. "Time works well: Dynamic time warping based on time weighting for time series data mining." *Information Sciences* 547 (2021): 592-608.
[5] Batista, Gustavo EAPA, et al. "CID: an efficient complexity-invariant distance for time series." *Data Mining and Knowledge Discovery* 28 (2014): 634-669.

**Models of geophysical behaviors**

[Figure]

Figure 1 : Cross-validated **Chandler wobble** model of luni-solar torqued Euler equations

[Figure]

Figure 2: Cross-validated **QBO** model at 70hPa of luni-solar forced Laplace's Tidal Equations

[Figure]

Figure 3: Cross-validated **ENSO** NINO4 modelof luni-solar forced Laplace's Tidal Equations

[Figure]

Figure 4 : Cross-validated **AMO** model of luni-solar forced Laplace's Tidal Equations

[Figure]

Figure 5 : Cross-validated **PDO** model of luni-solar forced Laplace's Tidal Equations

---

## Author Comment (AC1)

We are grateful to the two reviewers for their detailed and thoughtful comments (Sebastian Scher and the anonymous reviewer) and the community comments from Paul Pukite on our manuscript "*Potential for Equation Discovery with AI in the Climate Sciences*". We do recognise the time it takes to undertake reviewing tasks.

We are pleased that, based on these reviewers, the ESD journal has asked us to propose how we will adjust and enhance the manuscript. Our suggested alterations are listed below, with the requests in black text and our replies in indented blue font. This document covers both reviewers' and the community's comments.

**Anonymous Reviewer #1 Verdict: Minor Revisions**

Overall, I find this submission to be a relevant contribution to the climate science community as it provides a comprehensive overview of ongoing research, outlines current road-blocks and specifically suggests promising approaches which were previously over-looked or unknown in the field. Therefore, I highly suggest to improve upon the clarity and structure of the abstract, introduction and conclusion. This is necessary, as I find the importance of this contribution sometimes gets lost in overly long and disjunct paragraphs and sentences. While, I like the designated re-iteration of the potential application examples in the conclusions, I suggest to improve these by focussing on the discussed ways of using equation discovery in each application. My detailed review can be found in the supplement.

> We are grateful that the manuscript is regarded as relevant to climate science and offers promising approaches.
>
> The paper attempts to cover a lot of ground by combining current knowledge of equation discovery methods with three potential applications. However, upon returning to the MS, we accept that the presentation can be improved in places describing such connections. We will (1) tighten all wordings in the Abstract and Introduction, (2) scan for long sentences and split them where appropriate, and (3) change the re-iteration of examples in the Discussion and Conclusions.
>
> The major change is that we will replace most of the current Discussion reiteration text with a new Figure 5 (please see below for the proposed new diagram), which clearly presents the dimensions of the discovered equations. This diagram will provide more visual focus on how equation discovery fits in with the three scientific problems.

SUMMARY: This interesting paper introduces the promising research field of AI-led equation discovery to climate science. The authors provide an extensive overview over previous and current statical methods including machine learning based approaches in the field of climate science. As a remedy to the current issues such as transparency of most fully data-driven approaches and computational limitations of physics-based numerical solutions, the authors suggest the application of "equation discovery" algorithms. These AI-based algorithm enable equation generation for unknown dynamical system as well as for systems with limited dynamical information. Focussing on symbolic regression methods and specifically the SINDy algorithm, the authors provide a comprehensive, understandable and detailed description of

the procedure of equation discovery. Lastly, the examples of potential applications, such as in atmospheric convection, carbon cycle parameters, and ocean feature modelling for assessing tipping point risks, further strengthen the author's conclusion and outline promising research avenues.

> We are glad our MS is seen as interesting. The reviewer supports our view that AI may help determine the underpinning dynamical systems where current process knowledge and, thus, equation representation are limited or unknown. We are pleased the three examples from atmospheric, land, and oceans offer a variety of potential applications.

RESPONSE: Overall, I find this submission to be a relevant contribution to the climate science community as it provides a comprehensive overview of ongoing research, outlines current road-blocks and specifically suggests promising approaches which were previously over-looked or unknown in the field. Therefore, I highly suggest to improve upon the clarity and structure of the abstract, introduction and conclusion. This is necessary, as I find the importance of this contribution sometimes gets lost in overly long and disjunct paragraphs and sentences. While, I like the designated re-iteration of the potential application examples in the conclusions, I suggest to improve these by focussing on the discussed ways of using equation discovery in each application. I address my concerns in details below and discuss further individual remarks.

> As noted above, we will improve clarity throughout the paper, including alterations to long sentences. A major structural change will be replacing the section where the three scientific applications are re-iterated with a new Figure 5 (shown below). This diagram will place much more emphasis on the grid and dimensions of the actual equation discovery, allowing more specific details of the intended AI approach. This will leave text on the scientific drivers to be more focussed in the Sections above the Discussion part.

1. Relevance (*Does the paper address relevant scientific questions within the scope of ESD*?) This manuscript puts forward a new avenue of potential machine learning based earth system model (EMS) research, by suggesting AI-based equation discovery. Based on three highly relevant potential application examples, the authors demonstrate the relevance and motivate the integration of the proposed research direction.

> Thank you.

2. Novelty (*Does the paper present novel concepts, ideas, tools, or data*?) While the idea of equation discovery is an established sub-domain of machine learning and the discussed methods long-standing, their application to earth system modelling is a novel idea, to the best of my knowledge.

> Thank you.

3. Substantial conclusions (*Are substantial conclusions reached*?) The authors clearly establish their conclusions regarding the capabilities and potential of the AI-led equation discovery, throughout the manuscript

> Thank you.

4. Clarity and Validity (*Are the scientific methods and assumptions valid and clearly outlined*?) While the manuscript does exhibit structural and literary weaknesses, the algorithms in section 2.2 as well as potential applications in Section 3.3. are relayed in details and in an understandable manner.

> We accept the general criticism about presentation in the more "non-technical" parts of the paper. We will address those issues carefully.

5. Support of the interpretations and conclusions (*Are the results sufficient to support the interpretations and conclusions*?) The presented potential applications sufficiently support and highlight the relevance of the presented methods for climate science and earth system modelling.

> Thank you.

6. Traceability of results (*Is the description of experiments and calculations sufficiently complete and precise to allow their reproduction by fellow scientists* ?) As this paper aims to be more of a perspectives/review paper the authors do not present new experiments and therefore do not require exact reproducibility.

> Noted.

7. Consistency of related work (*Do the authors give proper credit to related work and clearly indicate their own new/original contribution*?) The authors consistently provide necessary citations and clearly establish the reviewed material as well as their own contributions.

> Thank you. We hope the reference list gets a balance between issues of climate change and pointers to the newly emerging methods of "AI Equation Discovery".

8. Title (*Does the title clearly reflect the contents of the paper*?) The manuscript title fully aligns with the manuscript content.

9. Abstract (*Does the abstract provide a concise and complete summary*?) The abstract provides a full summary of the manuscript content. However, I highly recommend to improve structure and overall writing, as it is hard to read and difficult to follow, which does not reflect the relevance and value of this submission.

> We intend to rewrite the Abstract and Introduction into a much more focused, slightly shorter format. We will shorten sentences and recheck the logic flows clearly as we build up to the more technical details in the core of the manuscript.

10. Structure and Clarity (*Is the overall presentation well structured and clear*?) While I enjoyed the explanatory figures and graphs, especially introduction and discussion should be improved to further strengthen the value of this manuscript. Overly long sentences

alongside sometimes disjunct paragraphs and sentences make these sections hard to follow. Exceptions are Section 2.2, 3.1, and 3.2 which, while long were easy to read.

Please see the response directly above (point 9) and our new Figure 5 (below). We propose that this new figure replaces most of the repetition of the scientific applications in the Discussion, allowing a better focus on the dimensions and practicality of the AI algorithm implementation. (Caption under diagram).

[Figure]

**Figure 5: Schematic of the grid of discovered equations**. For our three illustrative examples, represented by different rows, the left-hand side shows the mesh of the original data within which AI methods may discover underlying equations. The right-hand side is the potential numerical mesh of such equations. For atmospheric convection, the original data comprises very high-resolution simulated meteorological variables. Each variable is depicted as different 3-D blocks (vertically) and at different times (horizontally). Derived equations characterising high-resolution convection would be

embedded in existing ESMs on coarser scales, as shown on the top right. Land carbon modelling has two stages. Initially, at specific locations of FLUXNET data (yellow marks), timeseries of variables related to land-atmosphere carbon exchange, such as Net Ecosystem Productivity (NEP) and meteorological variations, would be used to derive time-evolving ODEs. Computer vision methods then calibrate and extrapolate these equations to all locations using high-resolution Earth Observation, ready for placement in ESM land components. Large-scale oceanic circulation modelling would first simply spatially average key depth-independent quantities, $T_1$, $T_2$, $T_{sub}$, $h_1$, $h_2$ and $\tau$, and equations are then found that describe their evolution in time, yielding a reduced complexity set of ODEs. Not all data would be used in the initial training exercise to determine governing equation sets. As with most AI methods, the remaining data would be used to test algorithms, which would determine the performance of the proposed equations. Hence, the arrow right-to-left at the bottom of the diagram. In some instances, there may be repeated cycles around these arrows, with alternative sets of equations derived for consideration and appropriate methods selected to compare them (e.g. the Akaike Information Criterion, AIC, statistics).

11. Language (*Is the language fluent and precise*?) I do have some concerns regarding the clarity of the paper (see previous point).

Please see above.

12. Math (*Are mathematical formulae, symbols, abbreviations, and units correctly defined and used*?) In Section 2.2 and Section 3.3, most mathematical expression are well-defined and explained. I only have minor concerns, which I detailed below.

The technical points you raise below are noted and will be corrected for.

13. Possible Reduction (*Should any parts of the paper (text, formulae, figures, tables) be clarified, reduced, combined, or eliminated*?) The manuscript would profit from a more precise and reduced section 2.1. (Background), discussion and introduction. In addition these parts should also be rewritten to improve clarity and readability, which currently hampers the value of this interesting contribution.

We will sharpen (and shorten) the background Section 2.1, and the Introduction. New Figure 5 will replace much of the text in the Discussion and Conclusions.

14. Number and quality of references (*Are the number and quality of references appropriate*?) The authors clearly cite all relevant works and choose relevant works out of the respective fields. However, while SINDy and symbolic regression is a well-renowned, the field of equation discovery also extends to more novel and promising algorithms, e.g., neural operators (Lu et al. 2021, Cao et al. 2023)

We will cite these additional references in Section 2.1 (and Section 2.2).

15. Supplement (*Is the amount and quality of supplementary material appropriate*?) I find this paper to be fully self-contained and therefore see no need for supplementary material.

MINOR CONCERNS:

1. Disjunct sentences: I find some sentences to be very hard to read, e.g. (p. 21 l. 8-10) *"First, to calibrate…"* T

TECHNICAL CORRECTIONS:

1. Sec. 2.2: Please clarify the dimensionality of $\xi$ (l. 6 p.8) is it the same as $\xi_2$. In addition I think a further specification of y might be helpful (l.7 p.8), since apparently $y = \Theta\xi$?

    Yes, we will correct for this, and as suggested.

2. Sec. 3.3: Please add definition/descriptions of $\beta$ and $Tr0$, since they appear to not be defined.

    We will add these definitions and sorry for their current omission.

3. L. 13-15, p. 12: "A major concern…" -> This sentence is not understandable please check the sentence structure. Cao, Qianying, Somdatta Goswami, and George Em Karniadakis. "LNO: Laplace neural operator for solving differential equations." arXiv preprint arXiv:2303.10528 (2023). Lu, Lu, et al. "Learning nonlinear operators via DeepONet based on the universal approximation theorem of operators." Nature machine intelligence 3.3 (2021): 218-229.

    Apologies, typos entered here during the final edits. These will be resolved.

**Sebastian Scher, Reviewer #2**

**Overall:**

The topic of the paper is timely and highly interesting, and the ideas presented in the paper are definitely worth publishing. The introduction part and the method part are well written and give a good overview. The main issue I see with the paper in its current form is that while the intention of the paper is obviously to provide ideas for AI solutions, most of the example part (part 3) of the paper discusses existing solutions or problem settings, without reference to potentially new solutions. For example, all three figures that describe the examples do not make any reference to any proposed new solutions.

Having said that, the ideas are definitely there in the paper, but especially in part 3, they tend to drown in overly long discussions of the domains of the examples, instead on focusing on potential use of AI.

    First, we are pleased that the paper ideas "are definitely worth publishing". Returning now to the manuscript, we understand the criticisms raised. Our proposed approach to offering clearer information on how AI will support new solutions is via our new Figure 5 and related caption (please see the new diagram above). This will focus on the dimensions of any newly derived equations and how they link to original

data, shifting the emphasis more onto the methods than the specific physical problem.

We plan to use the diagram to replace much of the text in Section "Discussion and Conclusions," where the three case studies are reiterated. We note that some of your requests are similar to those of Reviewer 1 and our additional Community Comment.

Therefore, while there is nothing fundamentally wrong or flawed with the paper in its current form, I would strongly encourage the authors to restructure especially part 3. Some ideas:

- Make the text focus more on potential AI solutions

- Replace the existing figures – which have a lot of details on the domains – with conceptual figures. With this, readers would be at a single glance be able to spot your ideas

  We understand these two points, but we are slightly reluctant to replace the existing figures. Many papers exist on AI techniques, including a growing (although still small) subset on equation discovery, that we cite. Our novelty is trying to relate the discovery method to potential applications in the climate sciences, and we hope the existing diagrams provide incentives and support for doing that.

  However, we have taken seriously the idea of a "conceptual figure" that captures all ideas at a "single glance." This is our new Figure 5 (see above, response to the first reviewer), where the emphasis is on the grid specification of any newly derived equations.

**Other comments**

it might be worthy to briefly mention explainable AI as well, and what differentiates equation discovery from it.

  We will make sure a sentence to this effect is added in the Introduction.

p.5 maybe mention weather clustering, which is widely used

  Thank you. Yes, this would be useful as an additional example. We shall mention it, along with a related citation, to illustrate where AI has provided extra understanding.

part 3.1: it is a bit unclear to what the task of an AI part would be in detail. There is in my opinion too much detail of the physics around it, and too little detail on how to solve the problems using equation discovery. E.g., instead of fig.2, which shows a lot of details on a convective storm (something that is actually not the topic of this paper), I think it would be more beneficial to have a conceptual graphic showing how to use equation discovery in this context.

As noted above, we respectfully request the retention of the process-based diagrams. By directly illustrating a set of climate-related problems that still have uncertainty, we hope to generate enthusiasm that equation discovery methods may reveal equation sets that capture process behaviour. However, we do agree that the paper needs to circle back to specific details of AI methods. The request here is for a conceptual graphic, and this has strong similarities to the request of our first reviewer. We hope that our new Figure 5 (see above) goes some way towards answering these requests.

3.2: same as for the first example: a conceptual figure on what is actually attempted would be very valuable.

Please see the comments above.

p.15 L5-6: " ML-derived spatial aggregation is a form of technique known as computer vision". Computer vision is a very broad field, so this sentence is incorrect. I also do not understand what you exactly mean here with "ML-derived". Again, less focus on the state of the field (here carbon cycle modeling) and more focus on the actual AI ideas would be better.

We will tighten the wording. We will point to specific potential methods of extrapolating (from single-point equations found by comparison against FLUXNET data) to climate model grid box scale and with EO methods.

3.3 in this section, even more than in 3.1 and 3.2, there is too much focus on existing methods and models, and the ideas using AI get lost in it. For example, I do not see the need of the detailed discussion, including equations, of the simple ENSO model.

We will shorten this part, but we believe that seeing a contemporary set of equations for one of the proposed applications provides a real incentive to use other methods to test their validity. Here, the question is: would "Equation Discovery" find similar equation sets and/or with similar parameters? We agree that the reasoning is not well written at this point, and we will sharpen the text in the new version accordingly.

p.15 L 22-23 "However, the large computational time of such simulations maintains interest in faster summary models, mainly in the form of coupled Ordinary Differential Equations (ODEs)." This should be reformulated, as it is a bit confusing, since also the high resolution ESMs are based on coupled ODEs.

Agreed. This is a definitional point we need to make clearer.

p.22 L3: the word "threat" seems an odd choice here.

We will change the use of this word.

In the paper, it is mentioned a couple of times (e.g. p21 L26) that it is unclear whether AI-developed models, because of their statistical nature, can extrapolate beyond current forcings. In our paper on Lorenz models, we showed – albeit in a highly simplified setting – that AI models indeed can to some extent learn the influence of external forcing and extrapolate it (https://npg.copernicus.org/articles/26/381/2019/npg-26-381-2019.html). This does of course not necessarily generalize to more complex models, but it might be worth mentioning.

> This is a key paper, and we are sorry that we did not cite it in the original submission. We will reference this paper in the new manuscript version.

**General style:**

Abbreviations: this is clearly a matter of personal taste, but considering the interdisciplinary nature of this paper, I would suggest to reduce the use of abbreviations. For example, TP (Tipping Points) is an abbreviation that many readers might not be familiar with, and it is anyway used used only a couple of times. Therefore, it would make the paper easier to read if it simply spelled out every single time (when I encountered it in the conclusion section, it took me some time to remember it meant tipping point even though it was mentioned in the introduction).

> We will revisit all acronyms, and if they are only used a small number of times after the original definition, then we avoid their usage.

Citation: https://doi.org/10.5194/esd-2024-30-RC2

**Paul Pukite, Community Comment**

The paper "Potential for Equation Discovery with AI in the Climate Sciences" is a vital discussion topic for advancing climate research. It's clear that there are infinitely many more non-linear formulations than the linear set of possibilities that humans are comfortable with solving. Fluid dynamics a la Navier-Stokes by itself contains many non-linear elements that have not been completely explored due to a lack of ability to solve in a closed form. The paper suggests an important possible constraint to apply: "*For physical systems involving fluid flows where the underlying equations are known to be energy preserving, although also nonlinear*".

> Thank you for the additional comment. We are pleased that you find the paper a "vital discussion topic".

And that's where artificial neural networks and symbolic regression (i.e. equation discovery) comes into play. There are really few other alternatives outside of tedious human trial & error that are available to both (1) fully explore the combinatorial solution space and (2) incorporate numerical solvers to train the possible solutions to fit the available data using appropriate metrics for plausibility and precision.

We are willing to highlight more that the proposed AI methods are likely to be particularly effective in the presence of nonlinearity. This is allowed by neural networks, as suggested, but also by the inclusion of nonlinear terms in candidate components of any discovered equations.

The paper as is falls short on two fronts, one of which the authors' themselves highlight. The first can be remedied by citing the importance of cross-validation (CV) strategies. The success of machine learning is in part due to how CV can separate the wheat from the chaff in potential solutions. Yet, nowhere in the text is cross-validation mentioned, and this is a vital part of equation discovery, as an optimal CV algorithm+metric is necessary to isolate candidate solutions along a Pareto front of complexity (1/plausibility) vs precision. Neural networks can fit just about any curve, so CV approaches to equation discovery help to eliminate those that are the result of over-fitting. Suggest Ref [1] as a citation starting point.

We take this first point seriously, recognising that although obvious, there needs to be a re-iteration that all AI methods require data to be split into training, with the remaining part available for testing. This split is just as important for testing any AI-derived equation set. It was a mistake of ours that this was not made clearer. We also illustrate this with the feedback loop (right-to-left) at the bottom of the new proposed Figure 5 (see above).

The second front is based on the authors' statement "*It is relatively easy to set aspirations for implementing AI methods in climate science, rather than performing the analysis itself*". I read this as a call to just do it instead of dreaming it, or as the thespian philosopher Christopher Walken said: "If you want to learn how to build a house, then build a house. Don't ask anybody. Just build a house." The paper suggested "*We discuss the potential application of AI-led equation discovery to three Earth system components. In each example, there is presently a deficiency in understanding, causing uncertainty in the representation of processes by equations. Each application falls into one of three categories*. "

We generally agree with this sentiment. However, there is also a time and place for more "Perspective" papers that set out new avenues or show how combining disciplines may lead to advances.

Instead, I would recommend three Earth system components to evaluate: solid body, atmosphere (gas fluid), and ocean (liquid fluid). In our text Mathematical Geoenergy, P. Pukite, D. Coyne, D. Challou (Wiley/AGU, 2019), we describe novel equation-based models for the Earth's Chandler wobble (solid body), QBO (atmosphere),and ENSO (ocean). The original nonlinear models were derived from simplifying Euler equations of motion for the Chandler wobble, and Laplace's Tidal Equations, which are simplified Navier-Stokes, for QBO and ENSO. We attain excellent agreement against observations in each case, and this extends to other climate indices such as AMO and PDO. See Figures 1..X at the end of this review.

We will review your paper and likely cite it. We are aware of deriving simplified equation sets with traditional methods (e.g. nondimensionalisation). In circumstances where such simplified models already exist, equation discovery may

provide additional verification that they have the most predictive capability. We can consider making that point around the climate examples you give here. Thank you for the suggestion.

Over the past few years, I have tried various machine learning approaches including neural networks and symbolic regression to observe if they would "discover" the same equation solutions I had formulated and applied. First, it's clear that neural networks can't do the job as they train only on their own data-set as supplied, and so won't automatically pull in all the tidal time-series data available. This is the closed-world assumption (CWA) problem well-known in AI circles for years, see Ref [2]. Neural networks will fit the data, but it's all based on dreaming up patterns from the data instead of tracing it back to a non-linear modulation from an external forcing. Alas, that external data set doesn't exist in the training data, so it gets ignored.

We note this concern with the use of neural network methods.

The symbolic regression/equation discovery approaches do an arguably better job. Although they also suffer from the CWA problem, they can make up for it by creating symbolic expressions from their library of primitive mathematical operators to draw from, such as creating a tidal forcing from (1) the time base, (2) arbitrary constants, and (3) sinusoidal primitives such as sin() and cos(). So, in terms of results, the frequencies from tidal factors do emerge in a symbolic regression fit to QBO, yet they are not directly harmonically-related due to the intrinsic non-linearity of the equation solutions! Thus, they may easily get overlooked when the symbolic regression results are deconstructed, as it also requires knowledge of nonlinear signal processing concepts such as aliasing and side-banding. That's what I have found straightforwardly in the Chandler wobble and QBO results, and with more of a challenge in the oceanic indices such as ENSO. The symbolic regression tools that I have evaluated include Eureqa, PySR, and TuringBot, Ref [3].

Again, we will look at these examples and the associated references. An implicit point that the reviewer makes here is to be aware that equations discovered may only be valid for particular time scales or length scales (or inadvertently bring in effects not actually present in the physical system). A knowledge of signal processing effects can act as a warning to this and can guide parameters set in discovery algorithms such as PySINDy. We will note this.

And this reflects back on the importance of cross-validation approaches and the selection of correlation metrics, including those that have proved valuable in machine learning in the context of noise and uncertainty, such as dynamic time warping - Ref [4] and complexity-invariance distance - Ref [5]. The results of symbolic regression depend on the best metric for the data, as some may prove too stiff to emerge from a local optima.

We will certainly emphasise the importance of keeping back a substantial fraction of data for test purposes. This can include testing whether derived equations are too stiff (or indeed the opposite, of too influenced by noise-related variability).

I agree with the paper that the focus on statistical machine learning to model climate variation is misguided, as it is more evident that large scale behaviors that are the result of collective deterministic actions describe better the standing wave models of ENSO and QBO. These will show the detail and variety in waveforms captured by wave equations, not the smeared responses captured by statistical ensembles.

> Yes! Our main motivation is that process-based equations govern the climate system, and in some instances, they remain unknown. Using the specific branch of AI that is "Equation Discovery", the hope is to bring an AI method to help, and one that is beyond only statistical representation.

Moreover (and finally), it is difficult to get a new paradigm accepted in geophysics fields such as climate science unless the results are beyond reproach. The complete lack of controlled experiments to test novel equation-based models means that claims of excellent agreement are dealt with suspicion. It is costly in terms of money and time to wait years for predictive models to come true, so the hope is that cross-validation results can conclusively demonstrate a new equation formulation has merit.

> Yes, again, although we would only ever suggest adopting a particular equation set after extensive comparison against independent test data ("controlled experiments") we realise we have not made that point clear in the text. This omission will be corrected in any revised paper version.

Ref

[1] Sweet, L., C. Müller, M. Anand, and J. Zscheischler, 2023: Cross-Validation Strategy Impacts the Performance and Interpretation of Machine Learning Models. Artif. Intell. Earth Syst., 2, e230026, https://doi.org/10.1175/AIES-D-23-0026.1.
[2] Zhu, Fei, Shijie Ma, Zhen Cheng, Xu-Yao Zhang, Zhaoxiang Zhang, and Cheng-Lin Liu. "Open-world machine learning: A review and new outlooks." arXiv preprint arXiv:2403.01759 (2024).
[3] In my opinion, Eureqa did the best job but it has not been available for use since 2017 as it was sold to an AI firm for proprietary use. https://en.wikipedia.org/wiki/Eureqa , https://turingbotsoftware.com/ which is a attempted clone of Eureqa. https://github.com/MilesCranmer/PySR
[4] Li, Hailin. "Time works well: Dynamic time warping based on time weighting for time series data mining." Information Sciences 547 (2021): 592-608.
[5] Batista, Gustavo EAPA, et al. "CID: an efficient complexity-invariant distance for time series." Data Mining and Knowledge Discovery 28 (2014): 634-669.

> For brevity, we have not repeated the diagrams presented in the Community Comment (please see online).

---

## Author Response (AR1)

We are grateful to the two reviewers for their detailed and thoughtful comments (Sebastian Scher and the anonymous reviewer) and the community comments from Paul Pukite on our manuscript *"Potential for Equation Discovery with AI in the Climate Sciences"*. We do recognise the time it takes to undertake reviewing tasks. We also thank the ESD journal editors for their support.

We are pleased that, based on these reviewers, the ESD journal has asked us to generate a revised manuscript. Our responses are listed below, with the requests in black text and our replies in indented blue font. This document covers both reviewers' and the community's comments. We have also taken the opportunity to sweep the full paper, removing a small number of typos and improving points of clarity.

A check was also made of all formatting issues, and to the best of our knowledge, the paper is compatible with Earth System Dynamics. As requested, in addition to the below, all changes can also be seen in our file generated by "latex_diff".

**Anonymous Reviewer #1 Verdict: Minor Revisions**

Overall, I find this submission to be a relevant contribution to the climate science community as it provides a comprehensive overview of ongoing research, outlines current road-blocks and specifically suggests promising approaches which were previously over-looked or unknown in the field. Therefore, I highly suggest to improve upon the clarity and structure of the abstract, introduction and conclusion. This is necessary, as I find the importance of this contribution sometimes gets lost in overly long and disjunct paragraphs and sentences. While, I like the designated re-iteration of the potential application examples in the conclusions, I suggest to improve these by focussing on the discussed ways of using equation discovery in each application.  My detailed review can be found in the supplement.

> We are grateful that the manuscript is regarded as relevant to climate science and offers promising approaches.
>
> The paper attempts to cover a lot of ground by combining current knowledge of equation discovery methods with three potential applications. However, upon returning to the MS, we accept that the presentation can be improved in places describing such connections. We will (1) tighten all wordings in the Abstract and Introduction, (2) scan for long sentences and split them where appropriate, and (3) change the reiteration of examples in the Discussion and Conclusions.
>
> The major change is that we have now replaced most of the current Discussion reiteration text with a new Figure 5 (please see below for the new diagram), which clearly presents the dimensions of the discovered equations. This diagram is designed to provide more visual focus on how equation discovery fits in with the three scientific problems we present.

SUMMARY: This interesting paper introduces the promising research field of AI-led equation discovery to climate science. The authors provide an extensive overview over previous and current statical methods including machine learning based approaches in the field of climate science. As a remedy to the current issues such as transparency of most fully data-driven approaches and computational limitations of physics-based numerical solutions, the authors suggest the application of "equation discovery" algorithms. These AI-based algorithm enable equation generation for unknown dynamical system as well as for systems with limited dynamical information. Focussing on symbolic regression

methods and specifically the SINDy algorithm, the authors provide a comprehensive, understandable and detailed description of the procedure of equation discovery. Lastly, the examples of potential applications, such as in atmospheric convection, carbon cycle parameters, and ocean feature modelling for assessing tipping point risks, further strengthen the author's conclusion and outline promising research avenues.

> We are glad our MS is seen as interesting. The reviewer supports our view that AI may help determine the underpinning dynamical systems where current process knowledge and, thus, equation representation are limited or unknown. We are pleased the three examples from atmospheric, land, and oceans offer a variety of potential applications.

RESPONSE: Overall, I find this submission to be a relevant contribution to the climate science community as it provides a comprehensive overview of ongoing research, outlines current road-blocks and specifically suggests promising approaches which were previously over-looked or unknown in the field. Therefore, I highly suggest to improve upon the clarity and structure of the abstract, introduction and conclusion. This is necessary, as I find the importance of this contribution sometimes gets lost in overly long and disjunct paragraphs and sentences. While, I like the designated re-iteration of the potential application examples in the conclusions, I suggest to improve these by focussing on the discussed ways of using equation discovery in each application. I address my concerns in details below and discuss further individual remarks.

> As noted above, we have improved clarity throughout the paper, including alterations to long sentences (please see the "latex_diff" file in particular to show these adjustments). However, a major structural change has been replacing the section where the three scientific applications are re-iterated with a new Figure 5 (shown below). This diagram places much more emphasis on the grid and dimensions of the actual equation discovery, allowing more specific details of the intended AI approach. The description of the three scientific examples is now far more focussed in the Sections above the Discussion part.

1. Relevance (*Does the paper address relevant scientific questions within the scope of ESD*?) This manuscript puts forward a new avenue of potential machine learning based earth system model (EMS) research, by suggesting AI-based equation discovery. Based on three highly relevant potential application examples, the authors demonstrate the relevance and motivate the integration of the proposed research direction.

> Thank you.

2. Novelty (*Does the paper present novel concepts, ideas, tools, or data*?) While the idea of equation discovery is an established sub-domain of machine learning and the discussed methods long-standing, their application to earth system modelling is a novel idea, to the best of my knowledge.

> Thank you.

3. Substantial conclusions (*Are substantial conclusions reached*?) The authors clearly establish their conclusions regarding the capabilities and potential of the AI-led equation discovery, throughout the manuscript

> Thank you.

4. Clarity and Validity (*Are the scientific methods and assumptions valid and clearly outlined*?) While the manuscript does exhibit structural and literary weaknesses, the algorithms in section 2.2 as well as potential applications in Section 3.3. are relayed in details and in an understandable manner.

> We accept the general criticism about presentation in the more "non-technical" parts of the paper. The Abstract and Introduction are now rewritten to be short, with long sentences split, and a very careful check that the logic building to the main part of the paper is clearer.

5. Support of the interpretations and conclusions (*Are the results sufficient to support the interpretations and conclusions*?) The presented potential applications sufficiently support and highlight the relevance of the presented methods for climate science and earth system modelling.

> Thank you.

6. Traceability of results (*Is the description of experiments and calculations sufficiently complete and precise to allow their reproduction by fellow scientists* ?) As this paper aims to be more of a perspectives/review paper the authors do not present new experiments and therefore do not require exact reproducibility.

> Noted.

7. Consistency of related work (*Do the authors give proper credit to related work and clearly indicate their own new/original contribution*?) The authors consistently provide necessary citations and clearly establish the reviewed material as well as their own contributions.

> Thank you. We hope the reference list gets a balance between issues of climate change and pointers to the newly emerging methods of "AI Equation Discovery".

8. Title (*Does the title clearly reflect the contents of the paper*?) The manuscript title fully aligns with the manuscript content.

9. Abstract (*Does the abstract provide a concise and complete summary*?) The abstract provides a full summary of the manuscript content. However, I highly recommend to improve structure and overall writing, as it is hard to read and difficult to follow, which does not reflect the relevance and value of this submission.

> The Abstract and Introduction are now more focused and in a slightly shorter format. Sentences are shorter, and the lead into the main paper component is hopefully sharper and more consistent. Please see the "latex_diff" for these editorial changes.

Structure and Clarity (*Is the overall presentation well structured and clear*?) While I enjoyed the explanatory figures and graphs, especially introduction and discussion should be improved to further strengthen the value of this manuscript. Overly long sentences

alongside sometimes disjunct paragraphs and sentences make these sections hard to follow. Exceptions are Section 2.2, 3.1, and 3.2 which, while long were easy to read.

Our main change to the paper is to rewrite the Discussion completely (we have also, however, rewritten the Introduction and scanned it to split long sentences where necessary – see "latex_diff").

We have generated a new Figure 5 (below), and use this to instead replace what was quite extensive of the repetition of the scientific applications in the Discussion. The diagram enables a better focus on the dimensions and practicality of the AI algorithm implementation. (Caption under diagram).

[Figure]

**Figure 5: Schematic of the grid of discovered equations.** For our three illustrative examples, represented by different rows, the left-hand side shows the mesh of the original data within which AI methods may discover underlying equations. The right-hand side is the potential numerical mesh of such equations. For atmospheric convection, the original data comprises very high-resolution simulated meteorological variables. Each variable is depicted as different 3-D blocks (vertically) and at different times (horizontally). Derived equations characterising high-resolution convection would be

embedded in existing ESMs on coarser scales, as shown on the top right. Land carbon modelling has two stages. Initially, at specific locations of FLUXNET data (yellow marks), timeseries of variables related to land-atmosphere carbon exchange, such as Net Ecosystem Productivity (NEP) and meteorological variations, would be used to derive time-evolving ODEs. Computer vision methods then calibrate and extrapolate these equations to all locations using high-resolution Earth Observation, ready for placement in ESM land components. Large-scale oceanic circulation modelling would first simply spatially average key depth-independent quantities, $T_1$, $T_2$, $T_{sub}$, $h_1$, $h_2$ and $\tau$, and equations are then found that describe their evolution in time, yielding a reduced complexity set of ODEs. Not all data would be used in the initial training exercise to determine governing equation sets. As with most AI methods, the remaining data would be used to test algorithms, which would determine the performance of the proposed equations. Hence, the arrow right-to-left at the bottom of the diagram. In some instances, there may be repeated cycles around these arrows, with alternative sets of equations derived for consideration and appropriate methods selected to compare them (e.g. the Akaike Information Criterion, AIC, statistics).

As we have changed the text of the Discussion and Conclusions so extensively, we present this new section as screenshots below:

[revised manuscript text omitted]

10. Language (*Is the language fluent and precise*?) I do have some concerns regarding the clarity of the paper (see previous point).

    Please see above.

11. Math (*Are mathematical formulae, symbols, abbreviations, and units correctly defined and used*?) In Section 2.2 and Section 3.3, most mathematical expression are well-defined and explained. I only have minor concerns, which I detailed below.

    They have been corrected (please see "latex_diff" file).

12. Possible Reduction (*Should any parts of the paper (text, formulae, figures, tables) be clarified, reduced, combined, or eliminated*?) The manuscript would profit from a more precise and reduced section 2.1. (Background), discussion and introduction. In addition these parts should also be rewritten to improve clarity and readability, which currently hampers the value of this interesting contribution.

    We have sharpened the background Section 2.1, and the Introduction. New Figure 5 will replace much of the text in the Discussion and Conclusions.

13. Number and quality of references (*Are the number and quality of references appropriate*?) The authors clearly cite all relevant works and choose relevant works out of the respective fields. However, while SINDy and symbolic regression is a well-renowned, the field of equation discovery also extends to more novel and promising algorithms, e.g., neural operators (Lu et al. 2021, Cao et al. 2023)

    Thank you for point us to these two references. In the paper, we now write:

    "*Deep neural networks have the inherent capability to approximate nonlinear functions, and, in certain setups, can also accurately approximate nonlinear operators. For instance, the DeepONet model developed by Lu et al. (2021) can approximate a diverse range of nonlinear continuous operators from data such as integrals, as well as implicit operators that represent deterministic and stochastic differential equations.*"

    "*We also note a novel data-driven method for solving ODEs and PDEs rather than "discovering" them, as introduced by Cao et al. (2023). In Cao et al. (2023), the Laplace neural operator is utilized for solving differential equations that can account for non-periodic signals, unlike the more well-known Fourier neural operator. The Laplace neural operator is an alternative approach to the more traditional numerical-solvers and can be advantageous since it has the capability to rapidly approximate solutions over a wide range of parameter values and without the need for further training.*"

14. Supplement (*Is the amount and quality of supplementary material appropriate*?) I find this paper to be fully self-contained and therefore see no need for supplementary material.

   MINOR CONCERNS:

1. Disjunct sentences: I find some sentences to be very hard to read, e.g. (p. 21 l. 8-10) "*First, to calibrate…*" T

   TECHNICAL CORRECTIONS:

1. Sec. 2.2: Please clarify the dimensionality of $\xi$ (l. 6 p.8) is it the same as $\xi_2$. In addition I think a further

specification of y might be helpful (l.7 p.8), since apparently y = Θξ?

*Thank you for this. The paper is now amended as follows:*

*"where epsilon and ζ are the strengths of zonal and vertical advection respectively (model bifurcation parameters) and are both dimensionless quantities."*

*"For simplicity, looking at this regression problem for only one system variable, let y be a vector of data measurements (i.e. a column of X) where $y \in R^m$. The fitting procedure is then attempting to minimise the difference between y and Θξ since y = Θξ where $\Theta(X) \in R^{mn}$ and $\xi \in R^n$."*

2. Sec. 3.3: Please add definition/descriptions of β and Tr0, since they appear to not be defined.

*Thank you, this has now been corrected as:*

1) *"Tr0 is the temperature beneath the thermocline"*
2) *"The variable β ($Km^{-1} s^{-1}$) quantifies the strength of the influence of thermocline depth perturbations on SSTs."*

3. L. 13-15, p. 12: "A major concern…" -> This sentence is not understandable please check the sentence structure. Cao, Qianying, Somdatta Goswami, and George Em Karniadakis. "LNO: Laplace neural operator for solving differential equations." arXiv preprint arXiv:2303.10528 (2023). Lu, Lu, et al. "Learning nonlinear operators via DeepONet based on the universal approximation theorem of operators." Nature machine intelligence 3.3 (2021): 218-229.

*Apologies, typos entered and these have been resolved.*

**Sebastian Scher, Reviewer #2**

**Overall:**

The topic of the paper is timely and highly interesting, and the ideas presented in the paper are definitely worth publishing. The introduction part and the method part are well written and give a good overview. The main issue I see with the paper in its current form is that while the intention of the paper is obviously to provide ideas for AI solutions, most of the example part (part 3) of the paper discusses existing solutions or problem settings, without reference to potentially new solutions. For example, all three figures that describe the examples do not make any reference to any proposed new solutions.

Having said that, the ideas are definitely there in the paper, but especially in part 3, they tend to drown in overly long discussions of the domains of the examples, instead on focusing on potential use of AI.

*First, we are pleased that the paper ideas "are definitely worth publishing". Returning to the manuscript, we understand the criticisms raised. Our approach is to now offer clearer information on how AI will support new solutions, presented via our new Figure 5 and related caption (please see the new diagram above in our reply to reviewer 1). The new figure allows a focus on the dimensions of any newly derived equations and how they link to original data, shifting the emphasis more onto the methods than the specific physical problem.*

*The diagram replaces much of the text in Section "Discussion and Conclusions," where the three case studies are reiterated. We note that some of your requests are similar to those of Reviewer 1 and our additional Community Comment. Extra to Figure 5 and shown in our response to reviewer 1 outlined above, are screenshots of the short Discussion component.*

Therefore, while there is nothing fundamentally wrong or flawed with the paper in its current form, I would strongly encourage the authors to restructure especially part 3. Some ideas:

- Make the text focus more on potential AI solutions

  Please see response directly below.

- Replace the existing figures – which have a lot of details on the domains – with conceptual figures. With this, readers would be at a single glance be able to spot your ideas

  We understand these two points, but we are slightly reluctant to replace the existing figures. Many papers exist on AI techniques, including a growing (although still small) subset on equation discovery, that we cite. Our novelty is trying to relate the discovery method to potential applications in the climate sciences, and we hope the existing diagrams provide incentives and support for doing that.

  However, we have taken seriously the idea of a "conceptual figure" that captures all ideas at a "single glance." This is our new Figure 5 (see above, response to the first reviewer), where the emphasis is on the grid specification of any newly derived equations. We hope this gives a broader feel to suggested "AI solutions".

**Other comments**

it might be worthy to briefly mention explainable AI as well, and what differentiates equation discovery from it.

Based on this request, we have now added the following paragraph.

"*A field of AI already existing is that of explainable AI (Linardatos et al. (2021)). This approach defines a set of methods and techniques that provide accessible and understandable justifications for predictions with ML, which are often "black-box" models such as neural networks. However, we make an important distinction that AI-led equation discovery can be considered stronger, instead as a form of interpretable AI, due to its inherent ability to produce human-readable and interpretable mathematical expressions as outputs. By default, the equations themselves are generally explainable. However, in some cases where the generated equations are complex and unintuitive, explainable AI methods may be needed to make the expressions more comprehensible (Aldeia and De França (2021)).*"

*Linardatos, P., Papastefanopoulos, V., and Kotsiantis, S.: Explainable AI: A Review of Machine Learning Interpretability Methods, Entropy, 23, doi:10.3390/e23010018, 2021.*

*Aldeia, G. and De França, F.: Measuring feature importance of symbolic regression models using partial effects, doi:10.1145/3449639.3459302, 2021.*

p.5 maybe mention weather clustering, which is widely used

We have considered this, but it is a very large scientific area, and we were slightly worried this would detract from the specifics of convective events (which arguably is a type of clustering dependent on large-scale forcings). However, we have worked on that part of the paper and including adding a new reference related to mesoscale convective systems (MCS).

Maybee, B., Marsham, J. H., Klein, C. M., Parker, D. J., Barton, E. J., Taylor, C. M., Lewis, H., Sanchez, C., Jones, R. W., and Warner, J.: Wind Shear Effects in Convection–Permitting Models Influence MCS Rainfall and Forcing of Tropical Circulation, Geophys. Res. Lett., 35 51, e2024GL110 119, doi:10.1029/2024GL110119, 2024.

part 3.1: it is a bit unclear to what the task of an AI part would be in detail. There is in my opinion too much detail of the physics around it, and too little detail on how to solve the problems using equation discovery. E.g., instead of fig.2, which shows a lot of details on a convective storm (something that is actually not the topic of this paper), I think it would be more beneficial to have a conceptual graphic showing how to use equation discovery in this context.

> As noted above, we respectfully request the retention of the process-based diagrams. By directly illustrating a set of climate-related problems that still have uncertainty, we hope to generate enthusiasm that equation discovery methods may reveal equation sets that capture process behaviour. However, we do agree that the paper needs to circle back to specific details of AI methods. The request here is for a conceptual graphic, and this has strong similarities to the request of our first reviewer. We hope that our new Figure 5 (see above), its caption and embedding in a fully-revised discussion section goes some way towards answering these requests.

3.2: same as for the first example: a conceptual figure on what is actually attempted would be very valuable.

> Please see the comments above.

p.15 L5-6: ” ML-derived spatial aggregation is a form of technique known as computer vision”. Computer vision is a very broad field, so this sentence is incorrect. I also do not understand what you exactly mean here with “ML-derived”. Again, less focus on the state of the field (here carbon cycle modeling) and more focus on the actual AI ideas would be better.

> We have tightened the wording where the land-atmosphere $CO_2$ exchange idea is presented. We have made clearer how the equation discovery is at point sources (via FLUXNET data), but we do contend that any subsequent extrapolation to ESM-gridbox scale, via Earth Observering data, is a potential type of computer vision. (We now write in a slightly more vague way, though, as “*Such ML-derived spatial aggregation could be a form of technique known as computer vision*”)

> We have taken very seriously moving more to an emphasis on actual AI ideas, and as noted already, we do this predominantly via our new Figure 5 (please see above). This new figure sets out any revised numerical mesh for discovered equations and also makes it very clear (via yellow markers) that equation improvement will be a two-stage process from single-point FLUXNET towers followed by EO datasets.

3.3 in this section, even more than in 3.1 and 3.2, there is too much focus on existing methods and models, and the ideas using AI get lost in it. For example, I do not see the need of the detailed discussion, including equations, of the simple ENSO model.

> We have improved the oceanic component. We are keen to retain a description of existing models, as they provide an interesting comparison against AI-derived new equation sets. However, to make this reasoning clearer, and based on this reviewer request, we now write towards the end of the paper: “*Finally, newer AI-derived reduced complexity equations, drawn from data or ESMs, may reveal if current simpler models, such as the Timmermann ENSO model, continue to be appropriate or if alternative versions of oscillator models are more valid*”.

p.15 L 22-23 “However, the large computational time of such simulations maintains interest in faster summary models, mainly in the form of coupled Ordinary Differential Equations (ODEs).” This should be reformulated, as it is a bit confusing, since also the high resolution ESMs are based on coupled ODEs.

Strictly speaking, full-complexity ocean models are PDE-based. However, the spatial aggregation from PDEs to simpler ODEs, to generate valuable simpler models, is now additionally covered in the revised Discussion (with, in the context of this manuscript, how AI may find such ODEs). Please see above for the new discussion section text.

p.22 L3: the word "threat" seems an odd choice here.

We have removed the use of this word.

In the paper, it is mentioned a couple of times (e.g. p21 L26) that it is unclear whether AI- developed models, because of their statistical nature, can extrapolate beyond current forcings. In our paper on Lorenz models, we showed – albeit in a highly simplified setting – that AI models indeed can to some extent learn the influence of external forcing and extrapolate it (https://npg.copernicus.org/articles/26/381/2019/npg-26-381-2019.html). This does of course not necessarily generalize to more complex models, but it might be worth mentioning.

This is a key paper, and we are sorry that we did not cite it in the original submission. We now write *"An emphasis on equation development, and their inherent description of processes, allows moving on from the complaint that AI-developed models are purely statistical and may fail if extrapolated to make predictions for higher future GHG levels (although some capability of statistical AI methods to predict new forcings is noted by Scher and Messori (2019))."*

**General style:**

Abbreviations: this is clearly a matter of personal taste, but considering the interdisciplinary nature of this paper, I would suggest to reduce the use of abbreviations. For example, TP (Tipping Points) is an abbreviation that many readers might not be familiar with, and it is anyway used used only a couple of times. Therefore, it would make the paper easier to read if it simply spelled out every single time (when I encountered it in the conclusion section, it took me some time to remember it meant tipping point even though it was mentioned in the introduction).

We have revisited all acronyms, and if they are only used a small number of times after the original definition, then we avoid their usage. (Details in the "latex_diff" file).

**Paul Pukite, Community Comment**

The paper "Potential for Equation Discovery with AI in the Climate Sciences" is a vital discussion topic for advancing climate research. It's clear that there are infinitely many more non-linear formulations than the linear set of possibilities that humans are comfortable with solving. Fluid dynamics a la Navier-Stokes by itself contains many non-linear elements that have not been completely explored due to a lack of ability to solve in a closed form. The paper suggests an important possible constraint to apply: "*For physical systems involving fluid flows where the underlying equations are known to be energy preserving, although also nonlinear*".

Thank you for the additional comment. We are pleased that you find the paper a "vital discussion topic".

And that's where artificial neural networks and symbolic regression (i.e. equation discovery) comes into play. There are really few other alternatives outside of tedious human trial & error that are available to both (1) fully explore the combinatorial solution space and (2) incorporate numerical solvers to train the possible solutions to fit the available data using appropriate metrics for plausibility and precision.

*We have borne this comment in mind as we have revised the paper. The Section "Symbolic Regression Methods for Equation Discovery to Uncover Unknown Dynamics" covers in much detail neural networks and the newer symbolic regression. Please see the "latex_diff" file for where we have sharpened this part of the paper.*

The paper as is falls short on two fronts, one of which the authors' themselves highlight. The first can be remedied by citing the importance of cross-validation (CV) strategies. The success of machine learning is in part due to how CV can separate the wheat from the chaff in potential solutions. Yet, nowhere in the text is cross-validation mentioned, and this is a vital part of equation discovery, as an optimal CV algorithm+metric is necessary to isolate candidate solutions along a Pareto front of complexity (1/plausibility) vs precision. Neural networks can fit just about any curve, so CV approaches to equation discovery help to eliminate those that are the result of over-fitting. Suggest Ref [1] as a citation starting point.

*We have taken this first point seriously, recognising that although obvious, there needs to be a re-iteration that all AI methods require data to be split into training, with the remaining part available for testing. This split is just as important for testing any AI-derived equation set, and so now at multiple locations in the paper, we reiterate how equation discovery must be both trained and then tested on parts of the original datasets.*

*It was a mistake of ours that this was not made clearer.*

*We also illustrate this with the feedback loop (right-to-left) at the bottom of the new proposed Figure 5 (see above).*

The second front is based on the authors' statement "*It is relatively easy to set aspirations for implementing AI methods in climate science, rather than performing the analysis itself*". I read this as a call to just do it instead of dreaming it, or as the thespian philosopher Christopher Walken said: "If you want to learn how to build a house, then build a house.
Don't ask anybody. Just build a house." The paper suggested "*We discuss the potential application of AI-led equation discovery to three Earth system components. In each example, there is presently a deficiency in understanding, causing uncertainty in the representation of processes by equations. Each application falls into one of three categories.*"

*We generally agree with this sentiment. However, there is also a time and place for more "Perspective" papers that set out new avenues or show how combining disciplines may lead to advances. We really hope our paper encourages mapping this newer AI method (of equation discovery) over to climate research.*

Instead, I would recommend three Earth system components to evaluate: solid body, atmosphere (gas fluid), and ocean (liquid fluid). In our text Mathematical Geoenergy, P. Pukite, D. Coyne, D. Challou (Wiley/AGU, 2019), we describe novel equation-based models for the Earth's Chandler wobble (solid body), QBO (atmosphere),and ENSO (ocean). The original nonlinear models were derived from simplifying Euler equations of motion for the Chandler wobble, and Laplace's Tidal Equations, which are simplified Navier-Stokes, for QBO and ENSO. We attain excellent agreement against observations in each case, and this extends to other climate indices such as AMO and PDO. See Figures 1..X at the end of this review.

*We reviewed in detail your paper. We recognize that you are proposing new potential forcings for oceanic-atmospheric coupling that may have a dependence on lunar and*

solar forcing. We will bear this in mind for future research and see how the concept develops in the literature in general. We hope this is acceptable at this stage and in the context of our current paper format.

Over the past few years, I have tried various machine learning approaches including neural networks and symbolic regression to observe if they would "discover" the same equation solutions I had formulated and applied. First, it's clear that neural networks can't do the job as they train only on their own data-set as supplied, and so won't automatically pull in all the tidal time-series data available. This is the closed-world assumption (CWA) problem well- known in AI circles for years, see Ref [2]. Neural networks will fit the data, but it's all based on dreaming up patterns from the data instead of tracing it back to a non-linear modulation from an external forcing. Alas, that external data set doesn't exist in the training data, so it gets ignored.

We thank you for this point, and we have taken this seriously. We now write in the paper: "*We briefly mention an issue that can arise with AI methods, known as the "closed-world assumption" (Chen and Liu (2018)). This issue arises if not all relevant knowledge is contained within the available data forming the training dataset. This may lead to a situation where previously unseen dynamics not captured during an AI training period may be present in the data held for testing and is therefore not recognized by the model. AI Models operating with this assumption cannot update themselves with new information especially in open and dynamic environments, where new features in data continually appear*."

Chen, Z., Liu, Y., and Sun, H.: Physics-informed learning of governing equations from scarce data, Nat. Commun., 12, doi:10.1038/s41467-021-26434-1, 2021.

The symbolic regression/equation discovery approaches do an arguably better job. Although they also suffer from the CWA problem, they can make up for it by creating symbolic expressions from their library of primitive mathematical operators to draw from, such as creating a tidal forcing from (1) the time base, (2) arbitrary constants, and (3) sinusoidal primitives such as sin() and cos(). So, in terms of results, the frequencies from tidal factors do emerge in a symbolic regression fit to QBO, yet they are not directly harmonically-related due to the intrinsic non-linearity of the equation solutions! Thus, they may easily get overlooked when the symbolic regression results are deconstructed, as it also requires knowledge of nonlinear signal processing concepts such as aliasing and side-banding. That's what I have found straightforwardly in the Chandler wobble and QBO results, and with more of a challenge in the oceanic indices such as ENSO. The symbolic regression tools that I have evaluated include Eureqa, PySR, and TuringBot, Ref [3].

Please see our response above to proposed additional forcings to oceanic variability, and our added text in recognition of the "Closed World Assumption" (CWA)

And this reflects back on the importance of cross-validation approaches and the selection of correlation metrics, including those that have proved valuable in machine learning in the context of noise and uncertainty, such as dynamic time warping - Ref [4] and complexity-invariance distance - Ref [5]. The results of symbolic regression depend on the best metric for the data, as some may prove too stiff to emerge from a local optima.

We note again at various points through the manuscript that, although clear to those developing AI algorithms, the importance of splitting data into two components of "train" and "test". Please see the "latex_diff" file, as this is added at multiple points.

I agree with the paper that the focus on statistical machine learning to model climate variation is misguided, as it is more evident that large scale behaviors that are the result of collective deterministic actions describe better the standing wave models of ENSO and QBO. These will show the detail and variety in waveforms captured by wave equations, not the smeared responses captured by statistical ensembles.

> Yes! Our main motivation is that process-based equations govern the climate system, and in some instances, they remain unknown. Using the specific branch of AI that is "Equation Discovery", the hope is to bring an AI method to help, and one that is beyond only statistical representation.

Moreover (and finally), it is difficult to get a new paradigm accepted in geophysics fields such as climate science unless the results are beyond reproach. The complete lack of controlled experiments to test novel equation-based models means that claims of excellent agreement are dealt with suspicion. It is costly in terms of money and time to wait years for predictive models to come true, so the hope is that cross-validation results can conclusively demonstrate a new equation formulation has merit.

> Thank you. Yes, again, although we would only ever suggest adopting a particular equation set after extensive comparison against independent test data ("controlled experiments") we realise we did not make that point clear in the original text. This omission is corrected in the revised paper version, and critically, we also illustrate this with the two arrows at the bottom of our new Figure 5.

Ref

[1] Sweet, L., C. Müller, M. Anand, and J. Zscheischler, 2023: Cross-Validation Strategy Impacts the Performance and Interpretation of Machine Learning Models. Artif. Intell. Earth Syst., 2, e230026, https://doi.org/10.1175/AIES-D-23-0026.1.
[2] Zhu, Fei, Shijie Ma, Zhen Cheng, Xu-Yao Zhang, Zhaoxiang Zhang, and Cheng-Lin Liu. "Open-world machine learning: A review and new outlooks." arXiv preprint arXiv:2403.01759 (2024).
[3] In my opinion, Eureqa did the best job but it has not been available for use since 2017 as it was sold to an AI firm for proprietary use. https://en.wikipedia.org/wiki/Eureqa , https://turingbotsoftware.com/ which is a attempted clone of Eureqa. https://github.com/MilesCranmer/PySR
[4] Li, Hailin. "Time works well: Dynamic time warping based on time weighting for time series data mining." Information Sciences 547 (2021): 592-608.
[5] Batista, Gustavo EAPA, et al. "CID: an efficient complexity-invariant distance for time series." Data Mining and Knowledge Discovery 28 (2014): 634-669.

> For brevity, we have not repeated the diagrams presented in the Community Comment (please see online).

---

## Author Response (AR2)

Dear Dr Messori – thank you for considering our manuscript again and for your additional comments, repeated below.

Our main changes are in Section 3, where we have worked to reduce the word length describing the three scientific challenges (atmospheric convective modelling, land-atmosphere carbon cycle exchange, and large-scale oceanic circulations).

For all three issues, by tightening the wording and, in some instances, referring back more comprehensively to the literature, we have removed just over a full page. All these alterations can be seen in the uploaded "track changes" document.

We have also amended the Figure captions to refer more directly to the proposed AI-led equation discovery. Specifically, at the end of Figure 2 (that illustrates a convective event), we now add to the caption: "*A key possibility for AI is to derive equation sets, potentially with stochastic components, that broadly aggregate these complex processes to scales of order 100 km, and so appropriate for inclusion in ESMs*".

Then, for Figure 4, which illustrates very large-scale ocean-atmosphere couplings, we add to the caption: "*We suggest that AI-led equation discovery is well-positioned to investigate oceanic datasets, in order to determine if the simplified model presented here remains the most appropriate to maximally represent the ocean-atmosphere system at very large scales*"

We apologise about the poor wording and replace "*stronger*" with "*more useful*". We have also amended the code availability statement as requested.

We very much hope that the manuscript is now in a state to be accepted by ESD. Please contact me if there are, however, any further questions.

Thank you again for all of your help, time and support on our Perspective article.

With kind regards,

Chris Huntingford (chg@ceh.ac.uk) and on behalf of all co-authors

*Dear Authors,*

*Thank you for your detailed replies to the Reviewers, and for including the new Figure 5. Even following your revisions, the paper is still quite lengthy for a perspective, and I agree with the concern expressed in some of the reviewer comments that there are passages in Sect. 3 which go into a lot of details not directly related to the proposed AI equation discovery approaches. I appreciate the point you make in your replies in wanting to show a clear connection to physical applications, but this needs to be balanced with the perspective format and accessibility to a broad*

*readership. I would thus encourage you to make a further effort to make the text as concise and easily readable as possible in this section. I would also recommend clarifying for the readers how the figures relate to the topic of the paper. Right now the captions to Fig. 2 and Fig. 4 make no clear link to AI equation discovery.*

*Two additional minor suggestions:*

*ll. 24-26 Please review this sentence; I struggled to understand what the term of comparison for "stronger" was.*

*Code availability: Please review the statement to read "This is a perspective article".*

*Best Regards,*
*Gabriele Messori*